

# Characteristic determinant and Manakov triple for the double elliptic integrable system

A. Grekov[1*] and A. Zotov[2†]

**1** Physics Department, Stony Brook University, USA
**2** Steklov Mathematical Institute of Russian Academy of Sciences, Gubkina str. 8, 119991, Moscow, Russia

⋆ grekovandrew@mail.ru, † zotov@mi-ras.ru

## Abstract

Using the intertwining matrix of the IRF-Vertex correspondence we propose a determinant representation for the generating function of the commuting Hamiltonians of the double elliptic integrable system. More precisely, it is a ratio of the normally ordered determinants, which turns into a single determinant in the classical case. With its help we reproduce the recently suggested expression for the eigenvalues of the Hamiltonians for the dual to elliptic Ruijsenaars model. Next, we study the classical counterpart of our construction, which gives expression for the spectral curve and the corresponding *L*-matrix. This matrix is obtained explicitly as a weighted average of the Ruijsenaars and/or Sklyanin type Lax matrices with the weights as in the theta function series definition. By construction the *L*-matrix satisfies the Manakov triple representation instead of the Lax equation. Finally, we discuss the factorized structure of the *L*-matrix.

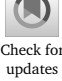

## Contents

**List of main notations:**

$q_j$, $j = 1, ..., N$ – positions of particles;

$\bar{q}_j = q_j - q_0$ – positions of particles in the center of mass frame, $q_0 = (1/N)\sum_k q_k$;

$q_{ij} = q_i - q_j$

$x_j = e^{q_j}$ or $x_j = e^{2\pi\iota q_j}$ in trigonometric or elliptic cases respectively;

$p_j$, $j = 1, ..., N$ – the classical momenta of particles;

$\omega = e^{2\pi\iota\tilde{\tau}}$ – the elliptic modular parameter, controlling the ellipticity in momenta;

$p = e^{2\pi\iota\tau}$ – the modular parameter, controlling the ellipticity in coordinates;

$q = e^{\hbar}$ – exponent of the Planck constant;

$t = e^{\eta}$ – exponent of the coupling constant;

$\lambda$ – the spectral parameter (1.1)   (sometimes called also $u$);

$z$ – the second spectral parameter (1.5);

$\mathcal{A}_{x,p}$ - the space of operators, generated by $\{x_1, .., x_N, q^{x_1\partial_1}, ..., q^{x_N\partial_N}\}$;

$:\quad:$ - normal ordering on $\mathcal{A}_{x,p}$, moving all shift of operators in each monomial to the right (2.25);

$\hat{\mathcal{O}}(\lambda)$ – the generating function of operators $\hat{\mathcal{O}}_n$ from [42] (1.1);

$\hat{\mathcal{O}}'(\lambda) = h^{-1}\hat{\mathcal{O}}(\lambda)h$ – the generating function $\hat{\mathcal{O}}(\lambda)$ with theta functions $\theta_p$ being replaced by the Jacobi theta functions $\vartheta$ (the function $h$ equals $\prod_{i<j} e^{-\pi\iota\eta(q_i-q_j)/\hbar}$, see (D.3))

$\hat{\mathcal{O}}'(z,\lambda)$ – extension of $\hat{\mathcal{O}}'(\lambda)$ by the second spectral parameter (2.26);

$\hat{H}(\lambda)$ – generating function of quantum Dell Hamiltonians $\hat{H}_n = \hat{\mathcal{O}}_0^{-1}\hat{\mathcal{O}}_n$ (1.2);

$\hat{\mathcal{H}}(\lambda)$ – alternative generating function of quantum Dell Hamiltonians $\hat{\mathcal{H}}_n = \hat{\mathcal{O}}^{-1}(1)\hat{\mathcal{O}}_n$ (5.1);

$\Xi \in \mathrm{Mat}(N,\mathbb{C})$ – the intertwining matrix, $\det\Xi$ is proportional to the Vandermonde function {9};

$g = \Xi D^{-1} \in \mathrm{Mat}(N,\mathbb{C})$ – the normalized intertwining matrix with diagonal matrix $D$ (6.10);

$\mathcal{L} = \sum_{n\in\mathbb{Z}} \omega^{\frac{n^2-n}{2}}(-\lambda)^n L^{RS}(q^n, t^n)$ – the weighted average of the Ruijsenaars Lax matrix (2.45);

$c$ – the light speed parameter as in the Ruijsenaars-Schneider model (5.14);

$L(z, \lambda) \in \mathrm{Mat}(N, \mathbb{C})$ – the $L$-matrix in the Manakov L-A-B triple (5.8).

All products of non-commuting operators should be understood as left ordered products. By this we mean:

$$\prod_{i=1}^{N} B_i = B_1 \cdot \ldots \cdot B_N \,.$$

# 1 Introduction and summary

## 1.1 Brief review

The double elliptic (or Dell) model [17–26] is an integrable system with an elliptic dependence on both – positions of particles and their momenta. It extends the widely known Calogero-Moser-Sutherland [27–31, 43] and Ruijsenaars-Schneider [60, 61] families of many-body integrable systems. Historically, the model was first derived as the elliptic self-dual system with respect to the Ruijsenaars (or equivalently, p-q or action-angle) duality interchanging positions of particles and action variables [57–59]. At the classical level the original group-theoretical Ruijsenaars construction was not applicable to the elliptic case. Instead, a geometrical approach was used based on the studies of spectral curves and Seiberg-Witten differentials [37]. In this way the Dell Hamiltonians where proposed in terms of higher genus theta-functions with a dynamical period matrices. For this reason a definition of the standard set of algebraic tools for integrable systems (including Lax pairs, $R$-matrix structures, exchange relations etc) appeared to be a complicated problem. The classical Poisson structures underlying the Dell model were studied in [2–4, 16].

An alternative version of the Dell Hamiltonians was suggested recently in [42]. The authors exploited the explicit form of the 6d Supersymmetric Yang-Mills partition functions with surface defects compactified on torus, which are conjectured to serve as the wavefunctions for the corresponding Seiberg-Witten intergable systems [1, 54–56]. The exact correspondence of their results with the previous studies is an interesting open problem though the matching has being already verified in a few simple cases. In this paper we deal with the Koroteev-Shakirov version of the generating function for commuting Hamiltonians. Namely, for the $N$-body system consider the operator of complex variables:

$$\hat{\mathcal{O}}(\lambda) = \sum_{n_1, \ldots, n_N \in \mathbb{Z}} \omega^{\sum_i \frac{n_i^2 - n_i}{2}} (-\lambda)^{\sum_i n_i} \prod_{i<j}^{N} \frac{\theta_p(t^{n_i - n_j} \frac{x_i}{x_j})}{\theta_p(\frac{x_i}{x_j})} \prod_i^{N} q^{n_i x_i \partial_i} = \sum_{n \in \mathbb{Z}} \lambda^n \hat{\mathcal{O}}_n \,. \tag{1.1}$$

This is a definition of the infinite set of (non-commuting) operators $\hat{\mathcal{O}}_k$. The positions of particles $q_i$ enter through $x_i = e^{q_i}$; $t = e^{\eta}$ – is exponent of the coupling constant $\eta$; $q = e^{\hbar}$ – is exponent of the Planck constant $\hbar$; and $\partial_i = \partial_{x_i}$, so that $\partial_{q_i} = x_i \partial_i$. The constant $\omega$ is the second modular parameter (controlling the ellipticity in momenta) and $\lambda$ is the (spectral) parameter of the generating function. The definition of the theta-function $\theta_p(x)$ with the constant modular parameter $\tau$ ($p = e^{2\pi i \tau}$) (controlling the ellipticity in coordinates) is given in (A.1). The commuting Hamiltonians of the Dell system were conjectured and argued to be of the form:

$$\hat{H}_n = \hat{\mathcal{O}}_0^{-1} \hat{\mathcal{O}}_n \,, \qquad n = 1, \ldots, N. \tag{1.2}$$

Solution to the eigenvalue problem for $\hat{H}_n$ was suggested in [5, 6] by extending the Shiraishi functions [64] – solutions to a non-stationary Macdonald-Ruijsenaars quantum problem.

Our study, on the contrary does not appeal to the explicit form of the wavefunctions and is mostly focused on the generating function itself. It is based on the usage of the intertwining matrix $\Xi(z)$ of the IRF-Vertex correspondence (see (9) for their explicit form) and the Hasegawa's factorization formula [39–41].

$$\hat{L}^{\text{RS}}(z, q, t) = g^{-1}(z) g(z - N\eta) q^{\text{diag}(\partial_{q_1}, \dots, \partial_{q_N})} \in \text{Mat}(N, \mathbb{C}) \ , \tag{1.3}$$

for the $\text{gl}_N$ elliptic Ruijsenaars-Schneider Lax operator with spectral parameter $z$ [60, 61]

$$\hat{L}^{RS}_{ij}(z, \eta, \hbar) = \frac{\vartheta(-\eta)\vartheta(z + q_{ij} - \eta)}{\vartheta(z)\vartheta(q_{ij} - \eta)} \prod_{k \neq j} \frac{\vartheta(q_{jk} + \eta)}{\vartheta(q_{jk})} e^{\hbar\partial_{q_j}} . \quad q_{ij} = q_i - q_j. \tag{1.4}$$

The matrix $\Xi(z) = \Xi(z, x_1, \dots, x_N | p)$ enters the normalized intertwining matrix $g(z, \tau) = \Xi(z) D^{-1}$ from (1.3), where $D(x_1, \dots, x_N)$ is a diagonal matrix used for convenient normalization only, see (6.10). A key property of these matrices, which will be used, is that $\det \Xi$ is proportional to the Vandermonde determinant. These intertwining matrices are known from the IRF-Vertex correspondence at quantum and classical levels [10–15, 44–47, 69]. The IRF-Vertex correspondence provides relation between dynamical and non-dynamical quantum (or classical) $R$-matrices as a special twisted gauge transformation with the matrix $g(z)$, thus relating the Lax operator (1.4) with the one of the Sklyanin type [65, 66].

## 1.2 Outline of the paper and summary of results

In this paper, using the Hamiltonians (1.1), we construct a generalization of the Macdonald determinant operator for the Dell system and study its applications.

We use a slightly modified and extended version of the generating function $\hat{\mathcal{O}}'(z, \lambda)$ (1.1), which depends on additional spectral parameter $z$, and generates an equivalent[1] set of operators $\hat{\mathcal{O}}'_k$:

$$\hat{\mathcal{O}}'(z, \lambda) = \sum_{k \in \mathbb{Z}} \frac{\vartheta(z - k\eta)}{\vartheta(z)} \lambda^k \hat{\mathcal{O}}'_k =$$

$$= \sum_{n_1, \dots, n_N \in \mathbb{Z}} \frac{\vartheta(z - \eta \sum_{i=1}^{N} n_i)}{\vartheta(z)} \omega^{\sum_i \frac{n_i^2 - n_i}{2}} (-\lambda)^{\sum_i^N n_i} \prod_{i < j}^N \frac{\vartheta(q_i - q_j + \eta(n_i - n_j))}{\vartheta(q_i - q_j)} \prod_i^N e^{n_i \hbar \partial_{q_i}} . \tag{1.5}$$

The paper is organized as follows.

In **Section 2** we derive the expression for the generalized Macdonald determinant:

$$\hat{\mathcal{O}}'(z - Nq_0, \lambda) = \frac{1}{\det \Xi(z)} \det_{1 \leq i, j \leq N} \left\{ \sum_{n \in \mathbb{Z}} (-\lambda)^n \omega^{\frac{n^2 - n}{2}} \Xi_i(q_j + n\eta, z) e^{n\hbar\partial_{q_j}} \right\}, \tag{1.6}$$

where $q_0$ is the center of mass coordinate. The determinant is well defined as the columns of the matrix commute. For the precise form of the matrix $\Xi_{ij} = \Xi_i(q_j, z)$ see (9).

In **Section 3** we express the generating function (1.5) in terms of the Lax matrix of the Ruijsenaars-Schneider model:

$$\hat{\mathcal{O}}'(z, \lambda) =: \det_{1 \leq i, j \leq N} \left\{ \hat{\mathcal{L}}^{\text{Dell}}_{ij}(z, \lambda | q, t | \tau, \omega) \right\} :, \tag{1.7}$$

where

$$\hat{\mathcal{L}}^{\text{Dell}}_{ij}(z, \lambda | q, t | \tau, \omega) = \sum_{k \in \mathbb{Z}} \omega^{\frac{k^2 - k}{2}} (-\lambda)^k \hat{L}^{RS}_{ij}(z | k\eta, k\hbar | \tau), \tag{1.8}$$

---

[1]Details of the relation between $\hat{\mathcal{O}}'_k$ and $\hat{\mathcal{O}}_k$ are given in (11).

and the normal ordering is defined in (2.25). The trigonometric and rational limits (for coordinate dependence) of (1.5)-(1.8) are described as well.

In **Section 4** we study the eigenvalue problem for the operator $\hat{\mathcal{O}}(u)$ (1.1) in the (coordinate) trigonometric limit $p = 0$, which corresponds to the dual to elliptic Ruijsenaars model[2], and compare our results to the known in the literature [5, 42].

The main statement here is the following: The operators $\hat{\mathcal{O}}(u)$ in the limit $p = 0$ for different $u$ could be simultaneously brought to the upper triangular form in some basis, their eigenvalues are labelled by Young diagrams $\lambda = (\lambda_1, ..., \lambda_N)$, and equal to:

$$E(u)_\lambda = \prod_{i=1}^{N} \theta_\omega(ut^{N-i}q^{\lambda_i}). \tag{1.9}$$

**In Section 5** we study the classical limit of the Dell system. Using the classical analogue $\mathcal{L}(z, \lambda)$ of (1.8) we show that the $L$-matrix

$$L(z, \lambda) = \mathcal{L}(z, 1)^{-1}\mathcal{L}(z, \lambda) \in \text{Mat}(N, \mathbb{C}), \tag{1.10}$$

satisfies the Manakov triple representation [34, 49] (instead of the Lax equation):

$$\dot{L} = [L, A] + BL, \quad \text{tr}B = 0. \tag{1.11}$$

The conservation laws are generated by the function $\det L(z, \lambda)$ only. It reduces to expression for the spectral curve of the Ruijsenaars-Schneider model in the $\omega \to 0$ limit.

In **Section 6** we describe the factorized structure for the $L$-matrix (1.10) $L(z, \lambda)$. Up to an inessential modification it is presented in the form, which is similar to the elliptic Kronecker function[3] (A.12):

$$\check{L}(z, \lambda | \tau, \tilde{\tau}) = \Phi[G(z, \tau), u | \tilde{\tau}] := \frac{\vartheta'(0 | \tilde{\tau})}{\vartheta(u | \tilde{\tau})}\Big[\vartheta(-\text{ad}_{N\eta\partial_z} | \tilde{\tau})G(z, \tau)\Big]^{-1}\vartheta(u - \text{ad}_{N\eta\partial_z} | \tilde{\tau})G(z, \tau),$$

$$u = \log(\lambda), \quad G(z, \tau) = g(z, \tau)\exp\Big(\frac{z}{Nc\eta}\text{diag}(p_1, ..., p_N)\Big) \in \text{Mat}(N, \mathbb{C}),$$
$$\tag{1.12}$$

thus generalizing the classical version of the factorization (1.3) to the double elliptic case. The elliptic moduli $\tilde{\tau}$ appears as $\omega = e^{2\pi\iota\tilde{\tau}}$. It is responsible for the ellipticity in momenta, while $\tau$ controls the ellipticity in positions of particles.

We also describe connection of the $L$-matrix with the Sklyanin Lax operators, and propose its quantization in terms of the elliptic quantum $R$-matrix in the fundamental representation of $\text{GL}_N$.

Possible applications of the obtained results and future plans are discussed in the end of the paper. Appendices contain the elliptic functions definitions and properties, description of the intertwining matrices $\Xi$, computations of $\text{GL}_2$ examples and relations between different forms of the generating functions.

## 2 Characteristic Macdonald determinant for the Dell system

In this Section we express the generating function $\hat{\mathcal{O}}(\lambda)$ (1.1) as a determinant of $N \times N$ matrix. The main idea is as follows. First, we introduce a certain bilinear pairing

---

[2]The terminology like dual to elliptic Ruijsenaars (or Calogero) model comes from the Mironov-Morozov description of the Dell model based on the p-q duality. Here and in what follows we use it meaning the trigonometric (or rational) $p = 0$ limit of (1.1), though its relation to p-q duality needs to be clarified.

[3]It is used in the widely known Lax pairs with spectral parameter [43, 60, 61] in many-body systems.

$\langle\ |\ \rangle : \mathcal{A}_x \times \mathcal{A}_p \to A_{x,p}$ between operators depending on coordinates and momenta only ($\mathcal{A}_x$ and $\mathcal{A}_p$ respectively). The generating function $\hat{\mathcal{O}}(\lambda)$ is then expressed as a pairing between the Vandermonde function and a product of theta functions of shift operators. Finally, we mention that the Vandermonde function is a part of determinant for some intertwining matrix $\Xi$. Using its properties we come to the determinant representation.

The set of intertwining matrices which we are going to use in different cases is given in the Appendix B . The elliptic coordinate case could not be treated without spectral parameter, while it is possible in the rational and trigonometric cases.

## 2.1 Determinant representation for the generating function of the Hamiltonians

### 2.1.1 The case without the spectral parameter

The result will be proven in all the details in the trigonometric case. The rational case could be done absolutely analogously. The main statement of this Section is the following.

**Theorem 2.1.** *Let $\Xi$ be the $N \times N$ Vandermonde matrix*

$$\Xi_{ij} = \Xi_i(x_j) = x_j^{N-i}, \tag{2.1}$$

*so*

$$\det_{1 \le i,j \le N} x_j^{N-i} = \prod_{i<j}(x_i - x_j). \tag{2.2}$$

*Then the generating function $\mathcal{O}^{trig}(\lambda)$ (1.1) in the coordinate trigonometric limit ($p = 0$)[4]:*

$$\hat{\mathcal{O}}^{trig}(\lambda) = \sum_{n_1,...,n_N \in \mathbb{Z}} \omega^{\sum_i \frac{n_i^2 - n_i}{2}} (-\lambda)^{\sum_i n_i} \prod_{i<j}^N \frac{t^{n_i} x_i - t^{n_j} x_j}{x_i - x_j} \prod_i^N q^{n_i x_i \partial_i}, \tag{2.3}$$

*is represented as follows:*

$$\hat{\mathcal{O}}^{trig}(\lambda) = \frac{1}{\prod_{i<j}(x_i - x_j)} \det_{1 \le i,j \le N} \left\{ \sum_{n \in \mathbb{Z}} (-\lambda)^n \omega^{\frac{n^2-n}{2}} \Xi_i(t^n x_j) q^{nx_j \partial_j} \right\}, \tag{2.4}$$

*or, equivalently,*

$$\hat{\mathcal{O}}^{trig}(\lambda) = \frac{1}{\prod_{i<j}(x_i - x_j)} \det_{1 \le i,j \le N} \left\{ x_j^{N-i} \theta_\omega(\lambda t^{N-i} q^{x_j \partial_j}) \right\}. \tag{2.5}$$

*Proof:* First, notice that the above determinant is well defined since any two elements from different columns of the corresponding matrix commute.

Consider the space of difference operators, generated by $\{x_1,...,x_N, q^{x_1 \partial_1},...,q^{x_N \partial_N}\}$. We refer to it as $\mathcal{A}_{x,p}$. Similarly, denote the spaces, generated by $\{x_1,...,x_N\}$ and $\{q^{x_1 \partial_1},...,q^{x_N \partial_N}\}$ only as $\mathcal{A}_x$ and $\mathcal{A}_p$ respectively.

Introduce the bilinear pairing:

$$\langle\ |\ \rangle\ : \mathcal{A}_x \times \mathcal{A}_p \to \mathcal{A}_{x,p}, \tag{2.6}$$

by defining it on the basis elements

$$\langle \prod_{i=1}^N x_i^{k_i} | \prod_{j=1}^N q^{m_j x_j \partial_j} \rangle = \prod_{l=1}^N t^{m_l k_l} \prod_{i=1}^N x_i^{k_i} \prod_{j=1}^N q^{m_j x_j \partial_j} \qquad \forall k_i, m_i. \tag{2.7}$$

---

[4]conjugated by $\prod_{i<j} x_j^{\frac{\ln t}{\ln q}}$

Due to $x_i q^{x_j \partial_j} = q^{x_j \partial_j} x_i$ for $i \neq j$, the pairing (2.7) satisfies an important property:

$$\langle \prod_{i=1}^{N} x_i^{k_i} | \prod_{j=1}^{N} q^{m_j x_j \partial_j} \rangle = \prod_{i=1}^{N} \langle x_i^{k_i} | q^{m_i x_i \partial_i} \rangle \qquad \forall\, k_i, m_i\,. \tag{2.8}$$

Then, the generating function (1.1) is represented as follows[5]:

$$\hat{\mathcal{O}}^{\text{trig}}(\lambda) = \frac{1}{\prod_{i<j}(x_i - x_j)} \langle \prod_{i<j}(x_i - x_j) | \prod_{k} \theta_\omega(\lambda q^{x_k \partial_k}) \rangle\,. \tag{2.9}$$

Next, we use the determinant property (2.2) and the linearity property of (2.7). From (2.9) we conclude

$$\hat{\mathcal{O}}^{\text{trig}}(\lambda) = \frac{1}{\prod_{i<j}(x_i - x_j)} \langle \det_{1 \leq i,j \leq N} \Xi_i(x_j) | \prod_{k} \theta_\omega(\lambda q^{x_k \partial_k}) \rangle = \tag{2.10}$$

$$= \frac{1}{\prod_{i<j}(x_i - x_j)} \sum_{\sigma \in S_N} (-1)^{|\sigma|} \langle \prod_{i} \Xi_{\sigma(i)}(x_i) | \prod_{k} \theta_\omega(\lambda q^{x_k \partial_k}) \rangle\,, \tag{2.11}$$

and the property (2.8) provides

$$\hat{\mathcal{O}}^{\text{trig}}(\lambda) = \frac{1}{\prod_{i<j}(x_i - x_j)} \sum_{\sigma \in S_N} (-1)^{|\sigma|} \sum_{n_1,\ldots,n_N \in \mathbb{Z}} (-\lambda)^{\sum_j n_j} \omega^{\sum_j \frac{n_j^2 - n_j}{2}} \langle \prod_{i} x_i^{N - \sigma(i)} | \prod_{k} q^{n_k x_k \partial_k} \rangle = \tag{2.12}$$

$$= \frac{1}{\prod_{i<j}(x_i - x_j)} \sum_{\sigma \in S_N} (-1)^{|\sigma|} \sum_{n_1,\ldots,n_N \in \mathbb{Z}} (-\lambda)^{\sum_j n_j} \omega^{\sum_j \frac{n_j^2 - n_j}{2}} \prod_{i} \langle x_i^{N - \sigma(i)} | q^{n_i x_i \partial_i} \rangle\,. \tag{2.13}$$

Hence,

$$\hat{\mathcal{O}}^{\text{trig}}(\lambda) = \frac{1}{\prod_{i<j}(x_i - x_j)} \sum_{\sigma \in S_N} (-1)^{|\sigma|} \prod_{i} \langle \Xi_{\sigma(i)}(x_i) | \theta_\omega(\lambda q^{x_i \partial_i}) \rangle\,. \tag{2.14}$$

Finally,

$$\hat{\mathcal{O}}^{\text{trig}}(\lambda) = \frac{1}{\prod_{i<j}(x_i - x_j)} \det_{1 \leq i,j \leq N} \langle \Xi_i(x_j) | \theta_\omega(\lambda q^{x_j \partial_j}) \rangle\,. \tag{2.15}$$

Expression under the determinant is easily calculated:

$$\langle \Xi_i(x_j) | \theta_\omega(\lambda q^{x_j \partial_j}) \rangle = \sum_{n \in \mathbb{Z}} (-\lambda)^n \omega^{\frac{n^2 - n}{2}} \Xi_i(t^n x_j) q^{n x_j \partial_j}\,. \tag{2.16}$$

This yields (2.4). Plugging the explicit expression for $\Xi$ into the r.h.s. of (2.16) and summing up over $n$ we get

$$\sum_{n \in \mathbb{Z}} (-\lambda)^n \omega^{\frac{n^2 - n}{2}} \Xi_i(t^n x_j) q^{n x_j \partial_j} = x_j^{N-i} \theta_\omega(\lambda t^{N-i} q^{x_j \partial_j})\,, \tag{2.17}$$

that is (2.5). ∎

Now let us write the answer for the rational case:

**Theorem 2.2.** *Let $\Xi$ be the $N \times N$ Vandermonde matrix*

$$\Xi_{ij} = \Xi_i(q_j) = (-q_j)^{i-1}\,, \tag{2.18}$$

---

[5]We knew this result from [63].

*so that*

$$\det_{1\leq i,j\leq N}(-q_j)^{i-1}=\prod_{i<j}(q_i-q_j).\tag{2.19}$$

*Then the generating function $\mathcal{O}(\lambda)$ (1.1) in the coordinate rational limit :*

$$\hat{\mathcal{O}}^{rat}(\lambda)=\sum_{n_1,\dots,n_N\in\mathbb{Z}}\omega^{\sum_i\frac{n_i^2-n_i}{2}}(-\lambda)^{\sum_i n_i}\prod_{i<j}^{N}\frac{q_i-q_j+\eta(n_i-n_j)}{q_i-q_j}\prod_{i}^{N}e^{n_i\hbar\partial_{q_i}},\tag{2.20}$$

*is represented as follows:*

$$\hat{\mathcal{O}}^{rat}(\lambda)=\frac{1}{\prod_{i<j}(q_i-q_j)}\det_{1\leq i,j\leq N}\left\{\sum_{n\in\mathbb{Z}}(-\lambda)^n\omega^{\frac{n^2-n}{2}}(-q_j-n\eta)^{i-1}e^{n\hbar\partial_{q_i}}\right\},\tag{2.21}$$

*or*

$$\hat{\mathcal{O}}^{rat}(\lambda)=\frac{1}{\prod_{i<j}(q_i-q_j)}\det_{1\leq i,j\leq N}\left\{\sum_{n\in\mathbb{Z}}(-\lambda)^n\omega^{\frac{n^2-n}{2}}\Xi_i(q_j+n\eta)e^{n\hbar\partial_{q_i}}\right\}.\tag{2.22}$$

*Proof:* The proof is word by word repetition of the trigonometric case. ∎

These theorems generalize the generating functions for commuting Hamiltonians for the quantum Calogero-Ruijsenaars family [39, 62].

### 2.1.2 The case with the spectral parameter

Let us proceed to the case with the spectral parameter. First of all we need to introduce a convenient notation. In this Section in place of the symbol $\theta_p(x)$ any of the three functions could be substituted

$$\theta_p\left(\frac{x_i}{x_j}\right)=\begin{cases}-(q_i-q_j)&\text{rational case,}\\1-\frac{x_i}{x_j}&\text{trigonometric case,}\\\theta_p\left(\frac{x_i}{x_j}\right)&\text{elliptic case.}\end{cases}\tag{2.23}$$

We will also use the odd theta function:

$$\vartheta(q_i-q_j)=\begin{cases}-(q_i-q_j)&\text{rational case,}\\e^{(q_i-q_j)/2}-e^{-(q_i-q_j)/2}&\text{trigonometric case,}\\\vartheta(q_i-q_j)&\text{elliptic case.}\end{cases}\tag{2.24}$$

For the precise relation between them, see (A.4) and (11).

In this Section, we will also need the normal ordering on $\mathcal{A}_{x,p}$, which moves all the shift operators to the right of all coordinates. Namely, on monomials

$$:\prod_{\mathcal{I}}x^{k_{\mathcal{I}}}q^{n_{\mathcal{I}}x\partial}:=\prod_{\mathcal{I}}x^{k_{\mathcal{I}}}\prod_{\mathcal{I}}q^{n_{\mathcal{I}}x\partial},\tag{2.25}$$

where $\mathcal{I}$ – multi-index.

Let us formulate the main statement. Define the new generating function:

$$\hat{\mathcal{O}}'(z,\lambda)=\sum_{k\in\mathbb{Z}}\frac{\vartheta(z-k\eta)}{\vartheta(z)}\lambda^k\hat{\mathcal{O}}'_k=$$

$$=\sum_{n_1,\dots,n_N\in\mathbb{Z}}\omega^{\sum_i\frac{n_i^2-n_i}{2}}(-\lambda)^{\sum_i^N n_i}\frac{\vartheta(z-\eta\sum_{i=1}^N n_i)}{\vartheta(z)}\prod_{i<j}^N\frac{\vartheta(q_i-q_j+\eta(n_i-n_j))}{\vartheta(q_i-q_j)}\prod_i^N e^{n_i\hbar\partial_{q_i}}.\tag{2.26}$$

Its relation to the previous one is also explained in (11).

**Theorem 2.3.** *Let $\Xi$ be the $N \times N$ matrix of the Stäckel form with the spectral parameter $z$*

$$\Xi_{ij} = \Xi_{ij}(\bar{q}_j, z) = \Xi_i\left(\bar{q}_j - \frac{z}{N}\right), \tag{2.27}$$

*with $\bar{q}_j$ being (B.5) and the property*

$$\det_{1 \le i,j \le N} \Xi_{ij}(\bar{q}_j, z) = c_N(\tau)\vartheta(z)\prod_{i<j}\vartheta(q_i - q_j). \tag{2.28}$$

*Then the generating function $\hat{\mathcal{O}}'(z, \lambda)$ (2.26) is represented as follows:*

$$\hat{\mathcal{O}}'(z, \lambda) = \frac{1}{\det \Xi_{ij}(\bar{q}_j, z)} : \det_{1 \le i,j \le N}\left\{\sum_{n \in \mathbb{Z}}(-\lambda)^n \omega^{\frac{n^2-n}{2}}\Xi_{ij}(\bar{q}_j + n\eta, z)e^{n\hbar\partial_{q_j}}\right\} :, \tag{2.29}$$

*or, equivalently,*

$$\hat{\mathcal{O}}'(z, \lambda) = \frac{1}{\det \Xi_{ij}(\bar{q}_j, z)} : \det_{1 \le i,j \le N}\left\{\sum_{k \in \mathbb{Z}}\Xi_{ij,k}(z)e^{(\alpha k+\sigma_{ij})q_j}\theta_\omega\left(\lambda e^{\alpha k\eta+\sigma_{ij}\eta}e^{\hbar\partial_{q_j}}\right)\right\} :, \tag{2.30}$$

*where the following expansions for the functions $\Xi_{ij}(\bar{q}_j, z)$ are assumed:*

$$\Xi_{ij}(\bar{q}_j, z) = \sum_{k \in \mathbb{Z}}\Xi_{ij,k}(z)e^{(\alpha k+\sigma_{ij})\bar{q}_j}, \tag{2.31}$$

*for some $\mathbb{C}$-numbers $\alpha$ and $\sigma_{ij}$.*

The explicit form of the matrices $\Xi$ is given in (B.6), (B.9) and (B.12). We use only these matrices in our study.

*Proof:* In order to use the trick from the Theorem 2.1 let us define the shifted $\Xi$ matrix, called $\tilde{\Xi}$:

$$\tilde{\Xi}_{ij} = \Xi_{ij}(q_j, z) = \Xi_{ij}(\bar{q}_j - q_0, z) = \Xi_{ij}(\bar{q}_j, z - Nq_0). \tag{2.32}$$

Its determinant is now equal to

$$\det_{1 \le i,j \le N}\tilde{\Xi}_{ij} = \det_{1 \le i,j \le N}\Xi_{ij}(q_j, z) = c_N(\tau)\vartheta(z - Nq_0)\prod_{i<j}\vartheta(q_i - q_j). \tag{2.33}$$

Now a matrix element $\tilde{\Xi}_{ij}$ depends on the coordinate $q_j$ only. Therefore, the following determinants

$$\frac{1}{\det \Xi_{ij}(q_j, z)} : \det_{1 \le i,j \le N}\left\{\sum_{n \in \mathbb{Z}}(-\lambda)^n \omega^{\frac{n^2-n}{2}}\Xi_{ij}(q_j + n\eta, z)e^{n\hbar\partial_{q_j}}\right\} :, \tag{2.34}$$

or

$$\frac{1}{\det \Xi_{ij}(q_j, z)} : \det_{1 \le i,j \le N}\left\{\sum_{k \in \mathbb{Z}}\Xi_{ij,k}(z)e^{(\alpha k+\sigma_{ij})q_j}\theta_\omega(\lambda e^{\alpha k\eta+\sigma_{ij}\eta}e^{\hbar\partial_{q_j}})\right\} :, \tag{2.35}$$

can be calculated as ordinary determinants since the elements from different columns commute. Indeed,

$$: \det_{1 \le i,j \le N}\left\{\sum_{n \in \mathbb{Z}}(-\lambda)^n \omega^{\frac{n^2-n}{2}}\Xi_{ij}(q_j + n\eta, z)e^{n\hbar\partial_{q_j}}\right\} :=$$

$$= \sum_{\sigma \in S_N}(-1)^{|\sigma|}\sum_{n_1,\dots,n_N \in \mathbb{Z}}\omega^{\sum_i \frac{n_i^2-n_i}{2}}(-\lambda)^{\sum_i^N n_i}\prod_{i=1}^N \Xi_{\sigma(i)i}(q_i + n_i\eta)\prod_{j=1}^N e^{n_j\hbar\partial_{q_j}} =$$

$$\tag{2.36}$$

$$= \sum_{\sigma \in S_N}(-1)^{|\sigma|}\sum_{n_1,\dots,n_N \in \mathbb{Z}}\omega^{\sum_i \frac{n_i^2-n_i}{2}}(-\lambda)^{\sum_i^N n_i}\prod_{i=1}^N \Xi_{\sigma(i)i}(q_i + n_i\eta)e^{n_i\hbar\partial_{q_i}} =$$

$$= \det_{1 \le i,j \le N}\left\{\sum_{n \in \mathbb{Z}}(-\lambda)^n \omega^{\frac{n^2-n}{2}}\Xi_{ij}(q_j + n\eta, z)e^{n\hbar\partial_{q_j}}\right\}.$$

Applying the pairing trick to them as in the proof of the Theorem 2.1, we arrive at

$$\frac{1}{\det \Xi_{ij}(q_j, z)} \langle \det_{1 \le i,j \le N} \Xi_{ij}(q_j, z) | \prod_k \theta_\omega(\lambda e^{\hbar \partial_{q_k}}) \rangle. \tag{2.37}$$

Substituting the explicit expression for the determinant of $\tilde{\Xi}$, one obtains

$$\frac{1}{\det \Xi_{ij}(q_j, z)} \langle c_N(\tau) \vartheta(z - Nq_0) \prod_{i<j} \vartheta(q_i - q_j) | \prod_k \theta_\omega(\lambda e^{\hbar \partial_{q_k}}) \rangle, \tag{2.38}$$

which, after taking the pairing equals

$$\sum_{n_1,\dots,n_N \in \mathbb{Z}} \omega^{\sum_i \frac{n_i^2 - n_i}{2}} (-\lambda)^{\sum_i^N n_i} \frac{\vartheta(z - Nq_0 - \eta \sum_{i=1}^N n_i)}{\vartheta(z - Nq_0)} \prod_{i<j}^N \frac{\vartheta(q_i - q_j + \eta(n_i - n_j))}{\vartheta(q_i - q_j)} \prod_i^N e^{n_i \hbar \partial_{q_i}}. \tag{2.39}$$

So, one obtains:

$$\hat{\mathcal{O}}'(z - Nq_0, \lambda) = \frac{1}{\det \Xi_i(q_j, z)} \det_{1 \le i,j \le N} \left\{ \sum_{n \in \mathbb{Z}} (-\lambda)^n \omega^{\frac{n^2-n}{2}} \Xi_i(q_j + n\eta, z) e^{n\hbar \partial_{q_j}} \right\}. \tag{2.40}$$

By the same argument as above, we could restore the normal ordering:

$$\hat{\mathcal{O}}'(z - Nq_0, \lambda) = \frac{1}{\det \Xi_i(q_j, z)} : \det_{1 \le i,j \le N} \left\{ \sum_{n \in \mathbb{Z}} (-\lambda)^n \omega^{\frac{n^2-n}{2}} \Xi_i(q_j + n\eta, z) e^{n\hbar \partial_{q_j}} \right\} :. \tag{2.41}$$

Finally, by shifting the parameter $z$ to $z + Nq_0$, we obtain the desired identity. ∎

## 2.2 Determinant representation in terms of the Ruijsenaars-Schneider $L$-matrix

In this paragraph we will derive one more useful representation for the generating function. Consider (2.4). Let us unify the Vandermonde determinant and the determinant of the sum into a single one. For this purpose we should put the normal ordering. From (2.4) we easily conclude

$$\hat{\mathcal{O}}^{\text{trig}}(\lambda) = \frac{1}{\prod_{i<j}(x_i - x_j)} \det_{1 \le i,j \le N} \left\{ \sum_{n \in \mathbb{Z}} (-\lambda)^n \omega^{\frac{n^2-n}{2}} \Xi_{ij}(t^n x_j) q^{nx_j \partial_j} \right\} =$$

$$=: \det_{1 \le i,j \le N} \left\{ \sum_{n \in \mathbb{Z}} \sum_{k=1}^N (-\lambda)^n \omega^{\frac{n^2-n}{2}} \Xi_{ik}^{-1} \Xi_{kj}(t^n x_j) q^{nx_j \partial_j} \right\} :. \tag{2.42}$$

The matrix (1.3):

$$\hat{L}^{RS}(q, t) = \sum_{k=1}^N \Xi_{ik}^{-1} \Xi_{kj}(tx_j) q^{x_j \partial_j}, \tag{2.43}$$

is the (quantum) trigonometric Ruijsenaars-Schneider Lax matrix. In order to rewrite it in its convenient form one should also perform the gauge transformation with the diagonal matrix $D_{ij} = \delta_{ij} \prod_{k \ne i}(x_i - x_k)$. See details in [40, 41, 69]. With the normal ordering the gauge transformed Lax operator has the same determinant. Hence, we arrive to the following determinant representation:

$$\hat{\mathcal{O}}^{\text{trig}}(\lambda) =: \det_{1 \le i,j \le N} \left\{ \sum_{n \in \mathbb{Z}} (-\lambda)^n \omega^{\frac{n^2-n}{2}} \hat{L}_{ij}^{RS}(t^n, q^n) \right\} =: \det \hat{\mathcal{L}}(\lambda) :, \tag{2.44}$$

where

$$\hat{\mathcal{L}}(\lambda) = \sum_{n \in \mathbb{Z}} (-\lambda)^n \omega^{\frac{n^2-n}{2}} \hat{L}^{RS}(t^n, q^n) \in \text{Mat}(N, \mathbb{C}), \qquad (2.45)$$

is the averaged sum of the Ruijsenaars Lax matrices. The averaging is over $\mathbb{Z}$ with the theta-function weights. Explicit form of $\hat{L}^{RS}(t, q)$ to be substituted into (2.45) is given in Section 3, expression (3.10). Its generation to the rational case is given by the expressions (3.13) and (3.14).

In the case with the spectral parameter the generating function depends on $z$. However, the arguments above could be repeated without any complications. Thus, its determinant representation is:

$$\hat{\mathcal{O}}'(z, \lambda) =: \det_{1 \le i,j \le N} \left\{ \sum_{n \in \mathbb{Z}} (-\lambda)^n \omega^{\frac{n^2-n}{2}} \hat{L}^{RS}_{ij}(z, q^n, t^n) \right\} :=: \det_{1 \le i,j \le N} \hat{\mathcal{L}}_{ij}(z, \lambda) :, \qquad (2.46)$$

$$\hat{\mathcal{L}}(z, \lambda) = \sum_{n \in \mathbb{Z}} (-\lambda)^n \omega^{\frac{n^2-n}{2}} \hat{L}^{RS}(z, q^n, t^n), \qquad (2.47)$$

where $\hat{L}^{RS}(z, q, t)$ is the elliptic Ruijsenaars-Schneider Lax matrix given by (3.2).

We present an alternative direct proof of the statements (2.44), (2.46) without usage of intertwining matrix in the next Section.

# 3 Alternative proof of relation to quantum Ruijsenaars-Schneider Lax operators

In this Section we give an alternative proof of the result from the paragraph (2.2), using the elliptic Cauchy determinant formula.

## 3.1 Double elliptic $\text{GL}_N$ model

The definition (1.1) can be alternatively written in terms of the standard odd Jacobi theta-function, see (11):

$$\hat{\mathcal{O}}'(\lambda) = \sum_{n_1, \dots, n_N \in \mathbb{Z}} \omega^{\sum_i \frac{n_i^2 - n_i}{2}} (-\lambda)^{\sum_i n_i} \prod_{i<j}^N \frac{\vartheta(q_i - q_j + \eta(n_i - n_j))}{\vartheta(q_i - q_j)} \prod_i^N e^{n_i \hbar \partial_{q_i}} = \sum_{n \in \mathbb{Z}} \lambda^n \hat{\mathcal{O}}'_n. \quad (3.1)$$

Therefore, the Hamiltonians $\hat{H}_n = (\hat{\mathcal{O}}'_0)^{-1} \hat{\mathcal{O}}'_n$ also commute. Its extension to the case with spectral parameter $z$ is given in (2.26).

We are in position to represent (2.26) in terms of the (quantum) elliptic Ruijsenaars-Schneider $\text{GL}_N$ Lax matrix with spectral parameter [39, 60, 61]:

$$\hat{L}^{RS}_{ij}(z, \eta, \hbar) = \frac{\vartheta(-\eta)\vartheta(z + q_{ij} - \eta)}{\vartheta(z)\vartheta(q_{ij} - \eta)} \prod_{k \ne j} \frac{\vartheta(q_{jk} + \eta)}{\vartheta(q_{jk})} e^{\hbar \partial_j}, \qquad (3.2)$$

where $q_{ij} = q_i - q_j$.

**Theorem 3.1.** *Let $\hat{L}^{RS}_{ij}(z, q, t)$ be the quantum Lax matrix for the elliptic Ruijsenaars-Schneider model (z - its spectral parameter), then the generating function (2.26) for $\hat{\mathcal{O}}'_n$ operators acquires the form:*

$$\hat{\mathcal{O}}'(z, \lambda) =: \det_{1 \le i,j \le N} \left\{ \sum_{n \in \mathbb{Z}} \omega^{\frac{n^2-n}{2}} (-\lambda)^n \hat{L}^{RS}_{ij}(z, n\eta, n\hbar) \right\} :=: \det_{1 \le i,j \le N} \hat{\mathcal{L}}_{ij}(z, \lambda) :, \qquad (3.3)$$

*with*

$$\hat{\mathcal{L}}_{ij}(z,\lambda) = \sum_{k\in\mathbb{Z}} \omega^{\frac{k^2-k}{2}}(-\lambda)^k \hat{L}_{ij}^{RS}(z,k\hbar,k\eta). \tag{3.4}$$

*Proof:* Consider the determinant

$$: \det\hat{\mathcal{L}} := \sum_\sigma (-1)^{|\sigma|} : \hat{\mathcal{L}}_{\sigma(1)1}\hat{\mathcal{L}}_{\sigma(2)2}\dots\hat{\mathcal{L}}_{\sigma(N)N} : \tag{3.5}$$

and substitute it into (3.3). Let us represent it as a sum of determinants. For this purpose collect all the terms with $\prod_i^N q^{n_i x_i \partial_i}$:

$$: \det\hat{\mathcal{L}} := \sum_{n_1,\dots,n_N\in\mathbb{Z}} \omega^{\sum_i \frac{n_i^2-n_i}{2}}(-\lambda)^{\sum_i n_i} \times$$

$$\times \sum_\sigma (-1)^{|\sigma|} : \hat{L}_{\sigma(1)1}^{RS}(z,n_1\eta,n_1\hbar)\hat{L}_{\sigma(2)2}^{RS}(z,n_2\eta,n_2\hbar)\dots\hat{L}_{\sigma(N)N}^{RS}(z,n_N\eta,n_N\hbar) : \tag{3.6}$$

$$= \sum_{n_1,\dots,n_N\in\mathbb{Z}} \omega^{\sum_i \frac{n_i^2-n_i}{2}}(-\lambda)^{\sum_i n_i} : \det_{1\le i,j\le N} \hat{L}_{ij}^{RS}(z,n_j\eta,n_j\hbar) :,$$

where the matrix $\hat{L}_{ij}^{RS}(z,n_j\eta,n_j\hbar)$ is constructed by combining rows from different terms of the sum (3.4). Using its explicit form (3.2) let us rewrite it through the elliptic Cauchy matrix:

$$\hat{L}_{ij}^{RS}(z,q_i-q_j,n_j\eta,n_j\hbar) = \vartheta(-n_j\eta)L_{ij}^{Cauchy}(z)\prod_{k:k\ne j}\frac{\vartheta(\tilde{q}_j-q_k)}{\vartheta(q_j-q_k)}e^{n_j\hbar\partial_j}, \tag{3.7}$$

where

$$L_{ij}^{Cauchy}(z) = \frac{\vartheta(z+q_i-\tilde{q}_j)}{\vartheta(z)\vartheta(q_i-\tilde{q}_j)}, \qquad \tilde{q}_j = q_j + n_j\eta. \tag{3.8}$$

Therefore,

$$: \det_{1\le i,j\le N} \hat{L}_{ij}^{RS}(z,q_i-q_j,n_j\eta,n_j\hbar) :=$$

$$\tag{3.9}$$

$$= \det_{1\le i,j\le N} L_{ij}^{Cauchy}(z)\prod_{k=1}^N\vartheta(-n_k\eta)\prod_{k,j:k\ne j}\frac{\vartheta(\tilde{q}_j-q_k)}{\vartheta(q_j-q_k)}\prod_{k=1}^N e^{n_k\hbar\partial_k}.$$

Plugging the Cauchy determinant (A.11) into (3.9) we get (2.26). ∎

The result of this Theorem is valid for the trigonometric and rational cases as well. The degenerations are obtained by substitutions $\vartheta(u) \to \sinh(u) \to u$.

Consider particular examples corresponding to the intertwining matrices (B.3), (B.9) and (B.1), (B.6). These are the models (p-q) dual to the elliptic Ruijsenaars-Schneider and elliptic Calogero-Moser models respectively[6].

## 3.2 Dual to the elliptic Ruijsenaars model

In the $GL_N$ case the relations (2.44)-(2.45) hold true for the Ruijsenaars-Schneider Lax matrix

$$\hat{L}_{ij}^{RS}(q,t) = \delta_{ij}\prod_{k\ne i}\frac{tx_i-x_k}{x_i-x_k}q^{x_i\partial_i} + (1-\delta_{ij})\frac{(1-t)x_j}{x_i-x_j}\prod_{k\ne i,j}\frac{tx_j-x_k}{x_j-x_k}q^{x_j\partial_j}. \tag{3.10}$$

---

[6]Let us notice once more that the terminology like dual to elliptic Ruijsenaars (or Calogero) model comes from the Mironov-Morozov description of the Dell model based on the p-q duality. Here and in what follows we mean the trigonometric (or rational) $p = 0$ limit of (1.1), though its relation to p-q duality needs to be clarified.

This expression corresponds to the intertwining matrix (B.3). Being substituted into (3.4) it gives the generating function (1.1) with $\theta_p(x) = 1 - x$ (up to conjugation).

The intertwining matrix (B.9) provides the Lax matrix with spectral parameter (see details in [69]):

$$\hat{L}_{ij}^{RS}(z) = -e^{-(N-2)\eta} \sinh(\eta) \Big( \coth(q_i - q_j - \eta) + \coth(z) \Big) \prod_{k:k\neq j}^{N} \frac{\sinh(q_j - q_k + \eta)}{\sinh(q_j - q_k)} e^{\hbar \partial_{q_j}} . \quad (3.11)$$

To get this Lax matrix one should substitute $y_j = e^{-2q_j + 2q_0 + 2z/N}$ into (B.10). Then from (B.11) we obtain the following generating function of the Hamiltonians related to the Lax matrix (3.11):

$$\hat{\mathcal{O}}'(z, \lambda) = \sum_{k \in \mathbb{Z}} \frac{\sinh(z - k\eta)}{\sinh(z)} \lambda^k \hat{\mathcal{O}}'_k = \sum_{n_1,\ldots,n_N \in \mathbb{Z}} \omega^{\sum_i \frac{n_i^2 - n_i}{2}} e^{(N-2)(z - \eta \sum_{i=1}^{N} n_i)} \times$$

$$\times \frac{\sinh(z - \eta \sum_{i=1}^{N} n_i)}{\sinh(z)} (-\lambda)^{\sum_i^N n_i} \prod_{i<j}^{N} \frac{\sinh(q_i - q_j + \eta(n_i - n_j))}{\sinh(q_i - q_j)} \prod_i^N e^{n_i \hbar \partial_{q_i}} . \quad (3.12)$$

## 3.3 Dual to the elliptic Calogero-Moser model.

Consider the $\Xi$ matrix (B.1), so that $\det \Xi_{ij} = \prod_{i<j}^{N} (q_i - q_j)$. The Lax matrix of the rational Ruijsenaars-Schneider model generated by this $\Xi$-matrix via (1.3) is of the form:

$$\hat{L}_{ij}^{RS}(q_i - q_j, \eta, \hbar) = \left( \frac{-\eta}{q_i - q_j - \eta} \right) \prod_{k:k\neq j}^{N} \frac{q_j - q_k + \eta}{q_j - q_k} e^{\hbar \partial_{q_j}} . \quad (3.13)$$

Being substituted into the averaged matrix

$$\hat{\mathcal{L}} = \sum_{k \in \mathbb{Z}} (-\lambda)^k \omega^{\frac{k^2 - k}{2}} \hat{L}^{RS}(k\eta, k\hbar) \in \mathrm{Mat}(N, \mathbb{C}), \quad (3.14)$$

it provides the following rational analogue for (1.1):

$$\hat{\mathcal{O}}'(\lambda) = \sum_{n_1,\ldots,n_N \in \mathbb{Z}} \omega^{\sum_i \frac{n_i^2 - n_i}{2}} (-\lambda)^{\sum_i n_i} \prod_{i<j}^{N} \frac{q_i - q_j + \eta(n_i - n_j)}{q_i - q_j} \prod_i^N e^{n_i \hbar \partial_{q_i}} . \quad (3.15)$$

In the case with spectral parameter $z$ we deal with intertwining matrix (B.6), which leads to the following Lax operator (see details in [69]):

$$\hat{L}_{ij}^{RS}(z, \eta, \hbar) = -\eta \left( \frac{1}{z} + \frac{1}{q_i - q_j - \eta} \right) \prod_{k:k\neq j}^{N} \frac{q_j - q_k + \eta}{q_j - q_k} e^{\hbar \partial_{q_j}} . \quad (3.16)$$

Then

$$\hat{\mathcal{O}}'(z, \lambda) = \sum_{n_1,\ldots,n_N \in \mathbb{Z}} \omega^{\sum_i \frac{n_i^2 - n_i}{2}} (-\lambda)^{\sum_i n_i} \left( \frac{z - \eta \sum_{i=1}^{N} n_i}{z} \right) \prod_{i<j}^{N} \frac{q_i - q_j + \eta(n_i - n_j)}{q_i - q_j} \prod_i^N e^{n_i \hbar \partial_{q_i}} . \quad (3.17)$$

The limit $z \to \infty$ of this answer yields the expression (3.15).

# 4 Eigenvalues for the dual to elliptic Ruijsenaars model

Let us now proceed to the first possible application of our result. We are going to derive the general formula for the eigenvalues of the dual to elliptic Ruijsenaars model. It corresponds to the trigonometric limit $p = 0$ of the Dell system. So, the generating function looks as follows:

$$\hat{\mathcal{O}}^{\text{trig}}(u) = \frac{1}{\prod_{i<j}(x_i - x_j)} \det_{1 \le i,j \le N} x_i^{N-j} \theta_\omega(ut^{N-j}q^{x_i \partial_i}). \tag{4.1}$$

By introducing standard notations $\delta_j = N - j$ and $\Delta$ for the Vandermonde determinant we write (4.1) as

$$\hat{\mathcal{O}}^{\text{trig}}(u) = \frac{1}{\Delta} \det_{1 \le i,j \le N} x_i^{\delta_j} \theta_\omega(ut^{\delta_j}q^{x_i \partial_i}), \qquad \Delta = \prod_{i<j}(x_i - x_j). \tag{4.2}$$

By the same arguments as for $\omega = 0$ case, we can see that the operator $\hat{\mathcal{O}}^{\text{trig}}(u)$ preserves the space $\Lambda_N$ of symmetric functions of the variables $x_1, ..., x_N$. So, we can consider the eigenvalue problem for the operator $\hat{\mathcal{O}}^{\text{trig}}(u)$:

$$\hat{\mathcal{O}}^{\text{trig}}(u)\Psi = E(u)\Psi, \tag{4.3}$$

where $\Psi$ is an element of $\Lambda_N$.

**Theorem 4.1.** *The eigenvalues of the operator $\hat{\mathcal{O}}^{\text{trig}}(u)$ are labeled by the Young diagrams $\lambda = (\lambda_1, ..., \lambda_N)$. The generating function of the eigenvalues takes the form:*

$$E(u)_\lambda = \prod_{i=1}^{N} \theta_\omega(ut^{N-i}q^{\lambda_i}). \tag{4.4}$$

*Proof:* Let $m_\lambda$ be the monomial symmetric function:

$$m_\lambda = \frac{1}{|S_\lambda|} \sum_{\sigma \in S_N} x^{\sigma(\lambda)} = \frac{1}{|S_\lambda|} \sum_{\sigma \in S_N} x_1^{\sigma(\lambda_1)} ... x_N^{\sigma(\lambda_N)}, \tag{4.5}$$

where $|S_\lambda|$ - the order of the symmetry group of the diagram $\lambda$, namely, the product of factorials of degeneracies of each row. They form a basis in the space of symmetric functions. Following the Macdonald's book [48] let us calculate their image under the action of the operator $\hat{\mathcal{O}}^{\text{trig}}(u)$:

$$\frac{1}{\Delta} \det_{1 \le i,j \le N} x_i^{\delta_j} \theta_\omega(ut^{\delta_j}q^{x_i \partial_i}) m_\lambda = \frac{1}{\Delta} \frac{1}{|S_\lambda|} \sum_{(\sigma, \sigma') \in S_N \times S_N} (-1)^{|\sigma|} \prod_{i=1}^{N} \theta_\omega(ut^{\sigma(\delta_i)}q^{\sigma'(\lambda_i)}) x_i^{\sigma'(\lambda_i)+\sigma(\delta_i)} \tag{4.6}$$

Next, make a change in the summation of variables by introducing $\pi$ with the property $\sigma' = \sigma \pi$. Then by rearranging the factors in the product we get

$$\hat{\mathcal{O}}^{\text{trig}}(u)m_\lambda = \frac{1}{\Delta} \frac{1}{|S_\lambda|} \sum_{(\pi, \sigma) \in S_N \times S_N} (-1)^{|\sigma|} \prod_{i=1}^{N} \theta_\omega(ut^{\delta_i}q^{\pi(\lambda_i)}) x_i^{\sigma(\pi(\lambda_i)+\delta_i)}, \tag{4.7}$$

or in terms of the Schur functions:

$$s_\lambda = \frac{1}{\Delta} \det(x_i^{\lambda_j + \delta_j}), \tag{4.8}$$

$$\hat{\mathcal{O}}(u)m_\lambda = \frac{1}{|S_\lambda|} \sum_{\pi \in S_N} \prod_{i=1}^{N} \theta_\omega(ut^{\delta_i}q^{\pi(\lambda_i)}) s_{\pi(\lambda)}. \tag{4.9}$$

Recall that the Schur functions have these properties:

 1) $s_{\pi(\lambda)}$ is either zero or equal to $\pm s_\mu$ for some $\mu < \lambda$ (unless $\pi(\lambda) = \lambda$);

 2) their relation to the monomial symmetric functions is given by

$$s_\lambda = m_\lambda + \sum_{\mu < \lambda} u_{\lambda\mu} m_\mu, \tag{4.10}$$

for some numbers $u_{\lambda\mu}$. Therefore, the operator $\hat{\mathcal{O}}(u)$ is upper triangular in the basis $\{m_\lambda\}$, and its eigenvalues have the form:

$$E(u)_\lambda = \prod_{i=1}^{N} \theta_\omega(ut^{N-i}q^{\lambda_i}) = \sum_{k\in\mathbb{Z}} u^k E_{k,\lambda}. \tag{4.11}$$

This finishes the proof. ■.

Hence, we could conclude that, despite not being commuting, the operators $\hat{\mathcal{O}}_n$ could be simultaneously brought to the upper triangular form in the basis $\{m_\lambda\}$. It would be interesting to find out, whether the analogous phenomenon takes place in the case $p \neq 0$.

### Eigenvalues in the $\mathrm{GL}_2$ case and comparison to the known answer

Let us write down the explicit expression for the eigenvalue $\mathcal{E}_{1,\lambda}$ of the first Hamiltonian (1.2) $\hat{H}_1 = \hat{\mathcal{O}}_0^{-1}\hat{\mathcal{O}}_1$. By expanding the result for the eigenvalues of $\hat{\mathcal{O}}(u)$ (4.11) in powers of $u$ (and to the first order in $\omega$) we obtain the eigenvalue of $\hat{H}_1$:

$$\mathcal{E}_{1,\lambda} = \frac{E_{1,\lambda}}{E_{0,\lambda}} = -\frac{\sum_i t^{N-i}q^{\lambda_i}}{1 + \omega \sum_{i\neq j} t^{-i+j}q^{\lambda_i-\lambda_j}} =$$
$$= -\sum_i t^{N-i}q^{\lambda_i} + \omega \sum_{i\neq j}\sum_k t^{N-i+j-k}q^{\lambda_i-\lambda_j+\lambda_k} + \dots. \tag{4.12}$$

For the $\mathrm{GL}_2$ case and the choice of $\lambda = (\lambda_1, \lambda_2) = (1,0)$ expression (4.12) yields

$$\mathcal{E}_{1,(1,0)} = -1 - tq + \omega\frac{(1+qt)(1+q^2t^2)}{qt} + \dots. \tag{4.13}$$

Up to the factor $t^{\frac{1}{2}}$ the results (4.11)-(4.13) coincide with those obtained in [42] and [5]. The factor $t^{\frac{1}{2}}$ comes from a slightly different definition of the Hamiltonians.

## 5   Classical mechanics: Manakov representation

In this section we will describe the classical limit of our construction and derive the Manakov *L-A-B* triple representation from it. The first step is to express the generating function of the Hamiltonians as the ratio of the two determinants. In the classical limit then, these two determinants could be combined into one, thus giving the expression for the classical spectral curve and the corresponding *L*-matrix.

### 5.1   One more generating function for the Dell Hamiltonians

So to fulfil the first step, described above, let us introduce an alternative version for the generating function of the commuting Dell Hamiltonians: put the operator $\hat{\mathcal{O}}(1) = \hat{\mathcal{O}}(\lambda)|_{\lambda=1}$ in (1.2) instead of $\hat{\mathcal{O}}_0$.

**Lemma 5.1.** *The operator*

$$\hat{\mathcal{H}}(\lambda) = \hat{\mathcal{O}}(1)^{-1}\hat{\mathcal{O}}(\lambda) = \sum_{n\in\mathbb{Z}} \lambda^n \hat{\mathcal{H}}_n, \quad \hat{\mathcal{H}}_n = \hat{\mathcal{O}}(1)^{-1}\hat{\mathcal{O}}_n, \tag{5.1}$$

*is also a generating function of the commuting Hamiltonians, so that commutativity of $\hat{\mathcal{H}}_n$ follows from commutativity of $\hat{H}_n$.*

*Proof:*   First, let us notice that the operators $\hat{H}_{kn} = \hat{\mathcal{O}}_k^{-1}\hat{\mathcal{O}}_n = \hat{H}_k^{-1}\hat{H}_n$ also commute with each other due to commutativity of $\hat{H}_k$. Therefore, $\hat{H}_{mk}\hat{H}_{nk} = \hat{H}_{nk}\hat{H}_{mk}$, or acting on this equality by $\hat{\mathcal{O}}_k^{-1}$ from the right

$$\hat{\mathcal{O}}_m^{-1}\hat{\mathcal{O}}_k\hat{\mathcal{O}}_n^{-1} = \hat{\mathcal{O}}_n^{-1}\hat{\mathcal{O}}_k\hat{\mathcal{O}}_m^{-1}. \tag{5.2}$$

Next, summing up over $k \in \mathbb{Z}$ gives

$$\hat{\mathcal{O}}_m^{-1}\hat{\mathcal{O}}(1)\hat{\mathcal{O}}_n^{-1} = \hat{\mathcal{O}}_n^{-1}\hat{\mathcal{O}}(1)\hat{\mathcal{O}}_m^{-1}. \tag{5.3}$$

By taking its inverse we get

$$\hat{\mathcal{O}}_n\hat{\mathcal{O}}(1)^{-1}\hat{\mathcal{O}}_m = \hat{\mathcal{O}}_m O(1)^{-1}\hat{\mathcal{O}}_n. \tag{5.4}$$

Finally, multiplying both sides by $\hat{\mathcal{O}}(1)^{-1}$ from the left yields

$$\hat{\mathcal{O}}(1)^{-1}\hat{\mathcal{O}}_n\hat{\mathcal{O}}(1)^{-1}\hat{\mathcal{O}}_m = \hat{\mathcal{O}}(1)^{-1}\hat{\mathcal{O}}_m\hat{\mathcal{O}}(1)^{-1}\hat{\mathcal{O}}_n, \tag{5.5}$$

for any $n$ and $m$, which is equivalent to $[\hat{\mathcal{H}}_n, \hat{\mathcal{H}}_m] = 0$. ∎

Due to (3.3) the generating function of the quantum Hamiltonians (5.1) takes the form:

$$\hat{\mathcal{H}}(\lambda) =: (\det_{1\leq i,j\leq N} \mathcal{L}_{ij}(z,1)):^{-1} : \det_{1\leq i,j\leq N} \mathcal{L}_{ij}(z,\lambda): . \tag{5.6}$$

On the one hand $\hat{\mathcal{O}}(1)$, compared to $\hat{\mathcal{O}}_0$ is hard to invert as its Taylor series expansion in $\omega$ starts not with 1. On the other hand the advantage of $\hat{\mathcal{O}}(1)$ is that it has determinant representation, while there is no natural way to find a determinant representation for $\hat{\mathcal{O}}_0$.

## 5.2 Spectral $L$-matrix

The operator $\hat{\mathcal{O}}(1)^{-1}$ in (5.1) really acts on $\hat{\mathcal{O}}(\lambda)$ as a quantum operator, so that we can not unify the normal orderings in (5.6). At the same time in the classical limit (5.6) reduces to

$$\mathcal{H}(z,\lambda) = \det_{N\times N}[\mathcal{L}^{-1}(z,1)\mathcal{L}(z,\lambda)], \tag{5.7}$$

that is, the matrix

$$L(z,\lambda) = \mathcal{L}^{-1}(z,1)\mathcal{L}(z,\lambda) \in \mathrm{Mat}(N,\mathbb{C}), \tag{5.8}$$

with

$$\mathcal{L}(z,\lambda) = \sum_{n\in\mathbb{Z}} (-\lambda)^n \omega^{\frac{n^2-n}{2}} L^{RS}(z,q^n,t^n), \tag{5.9}$$

arises, which determinant $\mathcal{H}(z,\lambda)$ is the generating function of the classical Hamiltonians. They commute with respect to canonical Poisson structure

$$\{p_i, q_j\} = \delta_{ij}. \tag{5.10}$$

Expression $\mathcal{H}(z,\lambda)$ can be considered as an analogue of the expression $\det(\lambda - l(z))$ for spectral curve of an integrable system with the Lax matrix $l(z)$. This is easy to see in the limit $\omega = 0$. Due to (A.5) we have

$$\mathcal{L}(z,\lambda)|_{\omega=0} = 1_N - \lambda L^{RS}(z,q,t), \tag{5.11}$$

where $1_N$ is the identity $N \times N$ matrix. Plugging (5.11) into (5.8) we get

$$L(z, \lambda) = \mathcal{L}^{-1}(z, 1)\mathcal{L}(z, \lambda) = \lambda 1_N + (1 - \lambda)\Big(1_N - L^{\mathrm{RS}}(z, q, t)\Big)^{-1}. \qquad (5.12)$$

Therefore, equation $\mathcal{H}(z, \lambda)|_{\omega=0} = 0$ is indeed the spectral curve of the elliptic Ruijsenaars-Schneider model (written in some complicated way). In the general case $L(z, \lambda)$ is not a Lax matrix. Its eigenvalues do not commute with respect to (5.10).

Let us remark that the existence of the Manakov's L-A-B triple does not contradict the possible simultaneous existence of some Lax pair. If we had a true Lax matrix for the Dell model then $\det L(z, \lambda)$ should represent its spectral curve. So, if the Lax representation exists we need to find a matrix $\check{L}$ of a size $M \times M$ (as was mentioned in [17–26] it is natural to expect $M = \infty$) and a change of variables $u = u(z, \lambda)$, $\zeta = \zeta(z, \lambda)$ satisfying

$$\det_{N \times N} \mathcal{L}(z, \lambda) = \det_{M \times M} \Big(u - \check{L}(\zeta)\Big). \qquad (5.13)$$

Another comment is about the geometrical meaning of $\mathcal{L}(z, \lambda)$ (5.9) and the $L$-matrix (5.8). In the Krichever-Hitchin approach to integrable systems these matrices are sections of (the Higgs) bundles over a base spectral curve with a coordinate $z$. The classical analogue of the Ruijsenaars-Schneider Lax matrix (1.4) takes the form

$$L_{ij}^{RS}(z) = \frac{\vartheta(-\eta)\vartheta(z + q_{ij} - \eta)}{\vartheta(z)\vartheta(q_{ij} - \eta)} e^{p_j/c} \prod_{k \neq j} \frac{\vartheta(q_{jk} + \eta)}{\vartheta(q_{jk})}, \quad q_{ij} = q_i - q_j. \qquad (5.14)$$

$c$ - the "speed of light" constant of the classical Ruijsenaars model. Its quasi-periodic behaviour is as follows: $L^{RS}(z + 1) = L^{RS}(z)$ and

$$L^{RS}(z + \tau) = e^{2\pi \iota \eta} \mathrm{Ad}_{\exp(-2\pi \iota \, \mathrm{diag}(q_1, \ldots, q_N))} L^{RS}(z). \qquad (5.15)$$

The first factor in (5.15) means that all terms in the sum (5.9) have different quasi-periodic behaviour on the lattice of periods $\langle 1, \tau \rangle$. Therefore, $\mathcal{L}(z, \lambda)$ is not a section of a bundle over the elliptic curve. This can be easily corrected by the substitution

$$\lambda = \lambda' \frac{\vartheta(z)}{\vartheta(z - \eta)}. \qquad (5.16)$$

Then the first factor in (5.15) get canceled, and we come to the quasi-periodic matrix $\mathcal{L}(z, \lambda)$. The $L$-matrix (5.8) is quasi-periodic as well. The price for this change of variables (5.16) is as follows. Initially we had the matrix $\mathcal{L}(z, \lambda)$ to be not quasi-periodic, but with a single simple pole at $z = 0$. After the change of variables we come to a quasi-periodic matrix function, but having higher order poles. The terms with positive $n$ in the sum (5.9) acquire the $n$-th order pole at $z = \eta$, and the terms with negative $n$ acquire the $(-n + 1)$-th order pole at $z = 0$. In what follows we do not use the substitution (5.16) keeping in mind that it can be done.

## 5.3 $L$-A-B **triple**

Consider the $L$-matrix (5.8) It is easy to see that this matrix satisfies identically the following equations known as the Manakov representation [49]:

$$\frac{d}{dt_k} L(z, \lambda) = [L(z, \lambda), M_k(z)] + B_k(z, \lambda) L(z, \lambda), \qquad (5.17)$$

where

$$\mathrm{tr} B_k(z, \lambda) = 0 \qquad (5.18)$$

and the "time" derivatives will be specified later. Indeed, by differentiating (5.8) we get

$$M_k(z) = \mathcal{L}^{-1}(z,\lambda)\Big(\frac{d}{dt_k}\mathcal{L}(z,\lambda)\Big), \tag{5.19}$$

so that the Manakov's $M_k(z)$ depends also on $\lambda$ in our case, and

$$B_k(z,\lambda) = \mathcal{L}^{-1}(z,\lambda)\Big(\frac{d}{dt_k}\mathcal{L}(z,\lambda)\Big) - \mathcal{L}^{-1}(z,1)\Big(\frac{d}{dt_k}\mathcal{L}(z,1)\Big). \tag{5.20}$$

The property (5.18) follows from

$$\frac{d}{dt_k}\det L(z,\lambda) = \frac{d}{dt_k}\frac{\det\mathcal{L}(z,\lambda)}{\det\mathcal{L}(z,1)}. \tag{5.21}$$

The l.h.s. of (5.21) equals zero since $\det L(z,\lambda) = \mathcal{H}(z,\lambda)$, while the r.h.s. is proportional to the trace of $B_k(z,\lambda)$ (5.20). Alternatively, introduce

$$M_k(z,\lambda) = \mathcal{L}^{-1}(z,\lambda)\Big(\frac{d}{dt_k}\mathcal{L}(z,\lambda)\Big). \tag{5.22}$$

Then it follows from the conservation of $\det L(z,\lambda)$ that

$$\mathrm{tr}M_k(z,\lambda) = \mathrm{tr}M_k(z,1). \tag{5.23}$$

Equations (5.17) are rather identities. To make them equivalent to the equations of motion one needs to replace the time derivative in the r.h.s. by its values following from the Hamiltonian equations of motion generated by some $k$-th Hamiltonian

$$\mathcal{H}_k(p,q) = \underset{z=0}{\mathrm{Res}}\,\underset{\lambda=0}{\mathrm{Res}}\Big(\frac{1}{z}\frac{1}{\lambda^{k+1}}\det L(z,\lambda)\Big). \tag{5.24}$$

Equations of motion are the standard Hamiltonian equations due to (5.10):

$$\frac{dq_i}{dt_k} = \{\mathcal{H}_k, q_i\} = \frac{\partial\mathcal{H}_k}{\partial p_i}, \quad \frac{dp_i}{dt_k} = \{\mathcal{H}_k, p_i\} = -\frac{\partial\mathcal{H}_k}{\partial q_i}. \tag{5.25}$$

Finally, by redefining (5.22) as

$$M_k(z,\lambda) = \mathcal{L}^{-1}(z,\lambda)\{\mathcal{L}(z,\lambda),\mathcal{H}_k\}, \tag{5.26}$$

we rewrite (5.19) and (5.20) as follows:

$$M_k(z) = M_k(z,\lambda), \tag{5.27}$$

and

$$B_k(z,\lambda) = M_k(z,\lambda) - M_k(z,1). \tag{5.28}$$

The expression $\{\mathcal{L}(z,\lambda),\mathcal{H}_k\}$ in (5.26) as well as the r.h.s. of equations of motion can be evaluated explicitly using the classical analogue of (1.2). With the definitions (5.27)-(5.28) the Manakov equations follow from the equations of motion.

Notice also that the Manakov equation (5.17) is easily rewritten as:

$$\frac{d}{dt_k}L(z,\lambda) = [L(z,\lambda), M_k(z,1)] + L(z,\lambda)B_k(z,\lambda), \tag{5.29}$$

or (what, in fact, directly follows from the definition of $L(z,\lambda)$)

$$\frac{d}{dt_k}L(z,\lambda) = L(z,\lambda)M_k(z,\lambda) - M_k(z,1)L(z,\lambda). \tag{5.30}$$

# 6 Factorization of the $L$-matrices

In this section we will show how our result naturally embeds the Dell model into the standard factorized Lax matrix approach, to the description of which we proceed in the next paragraph.

## 6.1 Classification of the factorized $L$-matrices

In [17–26] integrable many-body systems of the Calogero-Ruijsenaars family were naturally classified by the types of dependence on the coordinates and/or momenta. Both types are numerated by three possibilities. Each can be either rational, trigonometric or elliptic. For example, the choice (rational coordinate, trigonometric momenta) corresponds to the rational Ruijsenaars-Schneider model, while the choice (rational momenta, trigonometric coordinate) imply the trigonometric Calogero-Sutherland system. In the coordinate part this classification follows from solutions of the underlying functional equations [27–31,60,61] – the Fay identity (A.14) and its degenerations. By interchanging the types of coordinate and momenta dependence (as in the example pair above) one gets a pair of systems related by the Ruijsenaars duality transformation [57–59]. When both types coincide the corresponding model is self-dual. These are the rational Calogero-Moser system, the trigonometric Ruijsenaars-Schneider model, and finally the double elliptic model, which existence was predicted by these arguments.

Here we supply the upper classification with precise substitutions corresponding to the factorized Lax (or Manakov) $L$-matrices. As was discussed in [69] the factorized Lax matrices (with and without spectral parameter) for the systems of Calogero-Ruijsenaars type can be specified by a choice of two ingredients: the function $f$, and the intertwining matrix $\Xi(z)$:

$$L^{\mathrm{CR}}(z) = G^{-1}(z) f(-\mathrm{ad}_{N\eta\partial_z}) G(z), \qquad \mathrm{ad}_{\partial_z} * = [\partial_z, *], \tag{6.1}$$

where the matrix $G(z)$ is defined in terms of $\Xi(z)$:

$$G(z) = G(z, \tau) = \Xi(z) D^{-1} e^{\frac{z}{Nc\eta}P}, \qquad P = \mathrm{diag}(p_1, ..., p_N), \tag{6.2}$$

with some diagonal matrix $D$ (see (6.10) below) and $c$ – the light speed parameter[7] in the Ruijsenaars-Schneider model. The function $f(w)$ is either:

$$1) \ \text{linear:} \ f(w) = w; \tag{6.3}$$

$$2) \ \text{exponent:} \ f(w) = e^w. \tag{6.4}$$

The first choice of the function $f$ being substituted into (6.1) provides the Lax matrix of the Calogero-Moser-Sutherland systems [27–31,43]. The second choice of $f$ gives rise to the Lax matrices of the Ruijsenaars-Schneider models [60,61]. The choices of $\Xi(z)$ in (6.1) are given by (B.12) in the elliptic case, in the trigonometric case it is (B.9), and in the rational case it is (B.6). In trigonometric and rational cases one can also use the Vandermonde matrices (B.3) and (B.1) as $\Xi_{ij}(z) = (z - q_j)^{i-1}$ and $\Xi_{ij}(z) = (e^z x_j)^{N-i}$ respectively. The spectral parameter is cancelled out in these cases, and we get the Lax pairs of the Calogero-Ruijsenaars models without spectral parameter. See the review [69] for details.

Based on (1.1) and the Manakov $L$-matrix structure (5.8)-(5.9) we come to elliptic version for the function $f$:

$$3) \ \text{ratio of theta-functions:} \ f_\lambda(w) = \frac{\theta_\omega(\lambda e^w)}{\theta_\omega(e^w)}. \tag{6.5}$$

---

[7]The light speed provides non-relativistic limit $c \to \infty$ together with substitution $\eta = \nu/c$, where $\nu$ is the non-relativistic coupling constant.

The latter result is explained in the next paragraph in detail. A more universal classification picture arises if we slightly change the definition of $f_\lambda(w)$ as in transition from $\theta_p(e^w)$ to $\vartheta(w)$ (A.4) together with additional normalization factor $\vartheta'(0)/\vartheta(\log(\lambda))$. Then the function $f_\lambda(w)$ turns into the Kronecker elliptic function [70,71] depending on the moduli $\tilde{\tau}$ (defined through $\omega = e^{2\pi\iota\tilde{\tau}}$):

$$f_u(w) \to \lambda^{1/2} f_u(w) \frac{\vartheta'(0|\tilde{\tau})}{\vartheta(u|\tilde{\tau})} = \Phi(u, w|\tilde{\tau}) = \frac{\vartheta'(0|\tilde{\tau})\vartheta(w + u|\tilde{\tau})}{\vartheta(u|\tilde{\tau})\vartheta(w|\tilde{\tau})}, \tag{6.6}$$

where $u = \log(\lambda)$. In trigonometric and rational limits (when $\text{Im}(\tilde{\tau}) \to 0$) it is as follows:

$$\Phi^{\text{trig}}(u, w) = \frac{\sinh(w + u)}{\sinh(w)\sinh(u)} = \coth(w) + \coth(u),$$

$$\Phi^{\text{rat}}(u, w) = \frac{w + u}{wu} = \frac{1}{u} + \frac{1}{w}. \tag{6.7}$$

This function was used by I. Krichever [43] to construct the Lax representation with spectral parameter for elliptic Calogero-Moser system. It is widely used in elliptic integrable systems due to an addition Theorem known also as the genus one Fay identity (A.14). Considered as functional equation its solutions (including degenerated versions) were extensively studied [27–31].

In this way we come to the classification for the function $f_u(w)$ (responsible for the momenta type dependence), which is parallel to the well known classification of the coordinates dependence without spectral parameter [27–31] and with spectral parameter [43].

## 6.2 Factorized structure of the Dell $L$-matrix

Recall that the Lax matrix of the Ruijsenaars-Schneider model (3.2) is factorized as follows (1.3):

$$L^{\text{RS}}(z) = g^{-1}(z)g(z - N\eta)e^{P/c}, \tag{6.8}$$

where

$$g(z) = g(z, \tau) = \Xi(z)D^{-1}, \tag{6.9}$$

with the intertwining matrix $\Xi_{ij}(z)$ (B.12) and the diagonal matrix

$$D_{ij} = \delta_{ij}D_j = \delta_{ij}\prod_{k \neq j}\vartheta(q_j - q_k|\tau). \tag{6.10}$$

A conjugation with the latter diagonal matrix $D$ is performed in order to have a convenient form for $L^{\text{RS}}(z)$. Consider the matrix $G(z)$ (6.2). The Ruijsenaars-Schneider Lax matrix (6.8) takes the form

$$\tilde{L}^{\text{RS}}(z) = G^{-1}(z)\text{Ad}_{e^{-N\eta\partial_z}}G(z), \tag{6.11}$$

up to gauge transformation with the diagonal matrix $\exp\left(\frac{z}{Nc\eta}P\right)$:

$$\tilde{L}^{\text{RS}}(z) = G^{-1}(z)G(z - N\eta) = e^{-\frac{z}{Nc\eta}P}L^{\text{RS}}(z)e^{\frac{z}{Nc\eta}P}. \tag{6.12}$$

Let us proceed to the double elliptic case. Plugging (6.8) into the matrix $\mathcal{L}(z, \lambda)$ (5.9) we get

$$\mathcal{L}(z, \lambda) = g^{-1}(z)\sum_{k \in \mathbb{Z}}(-\lambda)^k \omega^{\frac{k^2-k}{2}}g(z - kN\eta)e^{kP/c}, \tag{6.13}$$

or

$$\mathcal{L}'(z, \lambda) = e^{-\frac{z}{Nc\eta}P}\mathcal{L}(z, \lambda)e^{\frac{z}{Nc\eta}P} = G^{-1}(z)\theta_\omega\left(\lambda\text{Ad}_{e^{-N\eta\partial_z}}\right)G(z). \tag{6.14}$$

By introducing also

$$\Theta(z,\lambda) = \theta_\omega\Big(\lambda \mathrm{Ad}_{e^{-N\eta\partial_z}}\Big) G(z) = \sum_{k\in\mathbb{Z}} (-\lambda)^k \omega^{\frac{k^2-k}{2}} g(z-kN\eta) e^{kP/c} e^{\frac{z}{Nc\eta}P} \in \mathrm{Mat}(N,\mathbb{C})\,, \quad (6.15)$$

we come to the following expression for the Manakov $L$-matrix (5.8):

$$L'(z,\lambda) = \Theta^{-1}(z,1)\Theta(z,\lambda)\,. \quad (6.16)$$

It is also gauge equivalent to both $\tilde{L}(z,\lambda)$ and $L(z,\lambda)$. In terms of the Kronecker function (6.6) we may write the Manakov $L$-matrix as

$$\check{L}(z,\lambda) = \Phi[G(z,\tau), u|\tilde{\tau}] := \frac{\vartheta'(0|\tilde{\tau})}{\vartheta(u|\tilde{\tau})}\Big[\vartheta(-\mathrm{ad}_{N\eta\partial_z}|\tilde{\tau})\,G(z)\Big]^{-1}\vartheta(u-\mathrm{ad}_{N\eta\partial_z}|\tilde{\tau})\,G(z)\,, \quad (6.17)$$

where $u = \log(\lambda)$. From the point of view of the classification of the factorized L-matrices discussed above, the expressions (6.15) and (6.17) can be considered as matrix analogues for theta-function and the Kronecker elliptic function respectively. The Kronecker function in (6.17) is constructed by means of the theta-operator $\Theta(z,\lambda)$ understood in a "plethystic sense" (6.15).

## 6.3 Relation to Sklyanin Lax operators

Due to the IRF-Vertex relation one can equivalently use the Sklyanin type Lax operators [65,66] instead of the Ruijsenaars-Schneider one in (1.8). Consider the gauge transformed Ruijsenaars Lax matrix (6.8):

$$L^{\mathrm{Skl}}(z) = g(z)L^{\mathrm{RS}}(z)g^{-1}(z) =$$
$$= g(z-N\eta)e^{P/c}g^{-1}(z) = \Xi(z-N\eta)e^{P/c}\Xi^{-1}(z)\,. \quad (6.18)$$

It is the classical analogue for the representation of the quantum Sklyanin Lax operator [40, 41]:

$$\hat{L}^{\mathrm{Skl}}(z) =: \Xi(z-N\eta)\,q^{\mathrm{diag}(\partial_{q_1},\dots,\partial_{q_N})/c}\,\Xi^{-1}(z) := \sum_{k=1}^{N}\Xi_{ik}(z-N\eta)\Xi_{kj}^{-1}(z)\,e^{(\hbar/c)\partial_{q_k}}\,. \quad (6.19)$$

Consider first the classical case (6.18). Its generalization to the double elliptic model is performed similarly to the previous paragraph. Define

$$\mathcal{L}^{\mathrm{Dell}}(z,\lambda) = g(z)\mathcal{L}(z,\lambda)g^{-1}(z) =$$
$$= \sum_{m\in\mathbb{Z}} (-\lambda)^m \omega^{\frac{m^2-m}{2}} \Xi(z-mN\eta) e^{mP/c} \Xi^{-1}(z)\,. \quad (6.20)$$

Hence,

$$\mathcal{L}^{\mathrm{Dell}}(z,\lambda) = \sum_{m\in\mathbb{Z}} (-\lambda)^m \omega^{\frac{m^2-m}{2}} L^{\mathrm{Skl}}(z,\{p_i\},\{q_i\},m\eta,mc^{-1})\,. \quad (6.21)$$

So, it could be expressed as a sum of the Sklyanin type Lax matrices with different coupling constants. For each of them, we know from [46,47] that it can be alternatively represented in terms of the underlying elliptic $R$-matrix:

$$L^{\mathrm{Skl}}(z,\eta,S(\{p_i\},\{q_i\},\eta,c^{-1})) = \sum_{a,b,c,d=1}^{N} E_{ab}S_{dc}(\{p_i\},\{q_i\},\eta,c^{-1})R_{ab,cd}(\eta,z)\,, \quad (6.22)$$

where $\{E_{ab}; a, b = 1, ..., N\}$ is the standard basis in $\mathrm{Mat}(N, \mathbb{C})$, the variables $S_{ab}(\{p_i\}, \{q_i\}, \eta, c^{-1})$ are generators of the classical Sklyanin algebra[8], and $R_{ab,cd}(\eta, z)$ – are the (properly normalized) weights of the quantum Baxter-Belavin $R$-matrix (in the fundamental representation of $\mathrm{GL}_N$):

$$R_{12}^{\eta}(z) = \sum_{a,b,c,d=1}^{N} E_{ab} \otimes E_{cd} R_{ab,cd}^{\mathrm{B}}(\eta, z). \tag{6.23}$$

With (6.22) we get the Manakov $L$-matrix

$$L^{\mathrm{Dell}}(z, \lambda) = \mathcal{L}^{\mathrm{Skl}}(z, 1)^{-1} \mathcal{L}^{\mathrm{Skl}}(z, \lambda), \tag{6.24}$$

with $\mathcal{L}^{\mathrm{Skl}}(z, \lambda)$ defined as

$$\mathcal{L}^{\mathrm{Skl}}(z, \lambda) = \sum_{k \in \mathbb{Z}} (-\lambda)^k \omega^{\frac{k^2-k}{2}} \sum_{a,b,c,d=1}^{N} E_{ab} S_{dc}(\{p_i\}, \{q_i\}, k\eta, kc^{-1}) R_{ab,cd}^{\mathrm{B}}(k\eta, z). \tag{6.25}$$

Expression (6.24) is gauge equivalent to the one (6.13) defined through the Ruijsenaars-Schneider Lax matrix.

**Quantization.** A natural quantization of (6.24), which gives the generating function (5.6) is as follows:

$$\hat{L}^{\mathrm{Dell}}(z, \lambda) = \left( : \hat{\mathcal{L}}^{\mathrm{Skl}}(z, 1) : \right)^{-1} : \hat{\mathcal{L}}^{\mathrm{Skl}}(z, \lambda) : . \tag{6.26}$$

Consider the matrix operator $: \hat{\mathcal{L}}^{\mathrm{Skl}}(z, \lambda) :$ (6.19). In the special case $\eta = -\hbar/c$ the Sklyanin Lax operator is represented as the quantum Baxter-Belavin $R$-matrix in the fundamental representation [7–9, 40, 41]. The $R$-matrix coefficients are of the form:

$$R_{ab,cd}^{\mathrm{B}}(\eta, z) = \delta_{a+c, b+d \bmod N} \frac{\theta^{(a-c)}(z+\eta)}{\theta^{(a-b)}(\eta)\theta^{(b-c)}(z)} \prod_{k=0}^{N} \frac{\theta^{(k)}(\eta)}{\theta^{(k)}(0)}, \tag{6.27}$$

where using the definition (A.6) we introduced

$$\theta^{(j)}(u) = \vartheta \begin{bmatrix} \frac{1}{2} - \frac{j}{N} \\ \frac{1}{2} \end{bmatrix} (u \,|\, N\tau). \tag{6.28}$$

In this way we come to the matrix representation for $\hat{L}^{\mathrm{Dell}}(z, \lambda)$ (6.26):

$$\hat{L}^{\mathrm{Dell}}(z, \lambda) = \mathbf{R}_{12}(z, \lambda) = \mathcal{R}_{12}(z, 1)^{-1} \mathcal{R}_{12}(z, \lambda) \in \mathrm{Mat}(N, \mathbb{C}), \tag{6.29}$$

with

$$\mathcal{R}_{12}(z, \lambda) = \sum_{a,b,c,d=1}^{N} E_{ab} \otimes E_{cd} \mathcal{R}_{ab,cd}^{\eta}(z, \lambda), \tag{6.30}$$

and

$$\mathcal{R}_{ab,cd}^{\eta}(z, \lambda) = \sum_{m \in \mathbb{Z}} (-\lambda)^m \omega^{\frac{m^2-m}{2}} R_{ab,cd}^{\mathrm{B}}(m\eta, z). \tag{6.31}$$

---

[8] The classical Lax matrix (6.22) describes integrable model of relativistic integrable top. Explicit change of variables with the Ruijsenaars model $S_{ab} = S_{ab}(\{p_i\}, \{q_i\}, \eta, c^{-1})$ appears in the case of rank one matrix $S$ (corresponding to special values of the Casimir functions). See details in [46, 47].

# 7 Discussion

- The Manakov $L$-matrix (6.24) can be used as a building block for the (partial) monodromy as it happens in integrable chains [34]. Namely, construct monodromy for a chain of length $L$:

$$T(z,\lambda) = L^{\text{Skl}}(z,\lambda,\{p_i^{(1)}\},\{q_i^{(1)}\}) L^{\text{Skl}}(z,\lambda,\{p_i^{(2)}\},\{q_i^{(2)}\})...L^{\text{Skl}}(z,\lambda,\{p_i^{(L)}\},\{q_i^{(L)}\}),$$
(7.1)

with the $L^{\text{Skl}}$ (6.24), and the $L$-matrix at each site depends on its own set of canonical variables. Then, $\det T(z,\lambda)$ provides a product of generating functions for each site. In [42] this was called the spin generalization of the double-elliptic model. Besides this construction one can also study the averaging of the spin Ruijsenaars-Schneider Lax operators.

- The quantization of (7.1) can be performed in a usual way

$$\mathbf{T}(z,\lambda) = \mathbf{R}_{01}(z,\lambda)\mathbf{R}_{02}(z,\lambda)...\mathbf{R}_{0L}(z,\lambda),$$
(7.2)

with $\mathbf{R}(z,\lambda)$ defined in (6.29)-(6.31). Presumably, a properly defined quantum determinant of $\mathbf{T}(z,\lambda)$ could be a generating function of commuting operators. Notice that, we did not prove the Yang-Baxter equation for $\mathbf{R}(z,\lambda)$ (it is rather not fulfilled since the traces of $T(z,\lambda)$ do not commute), so that $\mathbf{R}(z,\lambda)$ is not an $R$-matrix. Finding equation for $\mathbf{R}(z,\lambda)$ is another interesting problem.

- Orthogonality of the eigenvectors. To proceed further in our construction along with the analogous treatment for the usual Macdonald's symmetric functions we need to know the analog of the Macdonald's measure, with respect to which the operators $\hat{H}_n$ are self-conjugate. Probably, expressing the eigenvectors of $\hat{H}_n$ as vector-valued characters of some elliptic algebras might work, in the analogy with the paper [33].

- The integrable many-body systems can be also described via commuting differential or difference KZ-type connections or Dunkl operators using the Matsuo-Cherednik (or Heckman) projections respectively. The Ruijsenaars duality in many-body problems then turns (or rather embeds) into the spectral duality interchanging canonical coordinates and momenta being written in separated variables of the corresponding Gaudin models and/or spin chains. Much progress has been achieved in studies of these relations including its elliptic version [50–53]. An interesting problem is to find the Dunkl-Cherednik like description for the double-elliptic models (and define a double-elliptic version of the qKZ equations). We discuss these topics in our forthcoming paper [38].

- Let us mention a recent paper [67,68], where an elliptic integrable many-body system of Calogero type with the Manakov representation was obtained instead of the Lax representation. It was derived through a reduction from an integrable hierarchy. Presumably, it is a special limiting case, which can be deduced from the double-elliptic Manakov $L$-matrix.

- Further study of algebraic structures underlying the double-elliptic model is another set of important problems. This includes $r$-matrix structures at classical and quantum levels (RTT relations) and extensions of the Sklyanin quadratic algebras by means of $R$-matrix type operators (6.29)-(6.30).

- The determinant formula (1.7)-(1.8) can be naturally extended to the cases of many-body systems associated to the root systems of $BC_N$ type by substituting the Lax operators for the Ruijsenaars–Schneider–van Diejen type. This can be performed using results of [35,36]. We will discuss it in our future publication.

- Another open problem is to describe the classical $L$-matrix (5.8)-(5.9) in the group-theoretical (or Krichever-Hitchin) approach. As we mentioned in (5.16), one way is to consider the matrix valued function with higher order poles at a pair of marked points. Another possibility is to consider (block-matrix) direct sum $\bigoplus\limits_{k\in\mathbb{Z}} L^{\mathrm{RS}}(z, q^k, t^k)$ embedded into GL($\infty$). Each block is well defined, and the weighted sum of all blocks can be viewed as a matrix valued character.

  We will discuss these questions in future publications.

# 8 Appendix A: Elliptic functions

We use several different theta-functions. The first is the one used in [42]:

$$\theta_p(x) = \sum_{n\in\mathbb{Z}} p^{\frac{n^2-n}{2}}(-x)^n\,, \tag{A.1}$$

where the moduli of the elliptic curve $\tau\in\mathbb{C}$, Im $\tau > 0$ enters through

$$p = e^{2\pi i\tau}\,. \tag{A.2}$$

Another theta-function is the standard odd Jacobi one:

$$\vartheta(z) = \vartheta(z|\tau) = -i\sum_{k\in\mathbb{Z}}(-1)^k e^{\pi i(k+\frac{1}{2})^2\tau} e^{\pi i(2k+1)z}\,. \tag{A.3}$$

They are easily related:

$$\theta_p(x) = ip^{-\frac{1}{8}}x^{\frac{1}{2}}\vartheta(w|\tau)\,, \quad x = e^{2\pi iw}\,. \tag{A.4}$$

In the trigonometric limit $p\to 0$

$$\theta_p(x) \to (1-x)\,, \qquad \vartheta(w) \to -ip^{\frac{1}{8}}(\sqrt{x}-1/\sqrt{x})\,. \tag{A.5}$$

The Riemann theta-functions with characteristics are defined as follows:

$$\vartheta\!\left[\begin{array}{c} a \\ b \end{array}\right]\!(w|\tau) = \sum_{j\in\mathbb{Z}}\exp\left(2\pi\iota(j+a)^2\frac{\tau}{2} + 2\pi\iota(j+a)(w+b)\right)\,, \quad a,b\in\frac{1}{N}\mathbb{Z}\,. \tag{A.6}$$

In particular,

$$\vartheta\!\left[\begin{array}{c} 1/2 \\ 1/2 \end{array}\right]\!(w|\tau) = -\vartheta(w|\tau)\,. \tag{A.7}$$

For

$$V_{ij}(x_j) = \vartheta\!\left[\begin{array}{c} \frac{1}{2}-\frac{i}{N} \\ \frac{N}{2} \end{array}\right]\!(Nx_j\,|\,N\tau)\,, \tag{A.8}$$

we have

$$\det V = c_N(\tau)\vartheta(\sum_{k=1}^N x_k)\prod_{i<j}\vartheta(x_j-x_i)\,, \qquad c_N(\tau) = \frac{(-1)^{N-1}}{(\iota\,\eta_D(\tau))^{\frac{(N-1)(N-2)}{2}}}\,, \tag{A.9}$$

where $\eta_D(\tau)$ is the Dedekind eta-function:

$$\eta_D(\tau) = e^{\frac{\pi\iota\tau}{12}}\prod_{k=1}^{\infty}(1-e^{2\pi\iota\tau k})\,. \tag{A.10}$$

The determinant of the elliptic Cauchy matrix is given by

$$\det_{1\le i,j\le N} \frac{\vartheta(z+u_i-w_j)}{\vartheta(z)\vartheta(u_i-w_j)} = \frac{\vartheta\left(z+\sum_{i=1}^{N}(u_i-w_i)\right)}{\vartheta(z)} \frac{\prod_{p<q}^{N}\vartheta(u_p-u_q)\vartheta(w_q-w_p)}{\prod_{r,s=1}^{N}\vartheta(u_r-w_s)}.$$  (A.11)

Define the elliptic Kronecker function

$$\Phi(z,u) = \frac{\vartheta'(0)\vartheta(z+u)}{\vartheta(z)\vartheta(u)},$$  (A.12)

and the first Eisenstein function [70, 71]

$$E_1(z) = \frac{\vartheta'(z)}{\vartheta(z)}.$$  (A.13)

They satisfy the (genus one) Fay trisecant identity

$$\Phi(z,u_1)\Phi(w,u_2) = \Phi(z,u_1-u_2)\Phi(z+w,u_2) + \Phi(w,u_2-u_1)\Phi(z+w,u_1),$$  (A.14)

and its degeneration

$$\Phi(z,u)\Phi(w,u) = \Phi(z+w,u)(E_1(z)+E_1(w)+E_1(u)-E_1(z+w+u)).$$  (A.15)

Also,

$$E_1(z)+E_1(w)+E_1(u)-E_1(z+w+u) = \frac{\vartheta'(0)\vartheta(z+w)\vartheta(z+u)\vartheta(w+u)}{\vartheta(z)\vartheta(w)\vartheta(u)\vartheta(z+w+u)}.$$  (A.16)

# 9 Appendix B: Intertwining matrices

Following [69] we consider the intertwining matrices in two cases: with a spectral parameter and without a spectral parameter. These matrices lead to the Ruijsenaars-Schneider Lax matrices via (1.3) with and without spectral parameter respectively.

**The cases without spectral parameter.** Here we deal with the Vandermonde matrices in the rational and trigonometric coordinates.

1) in the rational case:

$$\Xi_{ij} = (-q_j)^{i-1},$$  (B.1)

$$\det\Xi = \prod_{i<j}(q_i-q_j).$$  (B.2)

2) in the trigonometric case:

$$\Xi_{ij} = x_j^{N-i}, \quad x_j = e^{q_j},$$  (B.3)

$$\det\Xi = \prod_{i<j}(x_i-x_j).$$  (B.4)

**The cases with a spectral parameter.** Here we deal with the Vandermonde type matrices, which are degenerated at $z = 0$. The special feature of these cases is that the simple zero at $z = 0$ appears in the center of mass frame only. Introduce

$$\bar{q}_j = q_j - q_0, \quad q_0 = \frac{1}{N}\sum_{k=1}^N q_k. \tag{B.5}$$

1) in the rational case:

$$\Xi_{ij}(z, q_j) = \left(\frac{z}{N} - q_j\right)^{\varrho(i)}, \tag{B.6}$$

with

$$\varrho(i) = \begin{cases} i-1 & \text{for } 1 \le i \le N-1, \\ N & \text{for } i = N. \end{cases} \tag{B.7}$$

Then

$$\det\Xi(z, q_j) = \left(z - \sum_{k=1}^N q_k\right)\prod_{i<j}(q_i - q_j), \qquad \det\Xi(z, \bar{q}_j) = z\prod_{i<j}(q_i - q_j), \tag{B.8}$$

2) in the trigonometric case we use

$$\Xi_{ij}(y_j) = y_j^{i-1} + \delta_{iN}\frac{(-1)^N}{y_j}, \tag{B.9}$$

$$\det\Xi = (-1)^{\frac{N(N-1)}{2}}\left(1 - \frac{1}{y_1 \dots y_N}\right)\prod_{i<j}(y_i - y_j). \tag{B.10}$$

Plugging $y_j = e^{-2q_j + 2q_0 + 2z/N}$ into (B.10) we get

$$\det\Xi = e^{(N-2)z}(e^z - e^{-z})\prod_{i<j}\left(e^{q_i-q_j} - e^{q_j-q_i}\right). \tag{B.11}$$

Notice that a possible difference in the Vandermonde determinant definitions including $\prod_{i<j}(x_i - x_j)$ or $\prod_{i<j}(1 - x_i/x_j)$ or $\prod_{i<j}(\sqrt{x_i/x_j} - \sqrt{x_j/x_i})$ can be easily removed through multiplying $\Xi$ by appropriate diagonal matrix. For example, in order to match the trigonometric Vandermonde function $\prod_{i<j}(x_i - x_j)$ to the trigonometric limit of (2.2) with $\theta_p(x_i/x_j) = 1 - x_i/x_j$ on should modify the trigonometric $\Xi$-matrix as $\Xi_{ij}(x_j) \to \Xi_{ij}(x_j)x_j^{\frac{N+1}{2} - j}$.

3) in the elliptic case

$$\Xi_{ij}(z, \bar{q}_j) = \vartheta\begin{bmatrix} \frac{1}{2} - \frac{i}{N} \\ \frac{N}{2} \end{bmatrix}\left(z - N\bar{q}_j \,|\, N\tau\right). \tag{B.12}$$

Due to (A.8) we obtain

$$\det\Xi(z, q) = c_N(\tau)\vartheta(z)\prod_{i<j}\vartheta(q_i - q_j). \tag{B.13}$$

## 10 Appendix C: Determinant representations in the $\mathrm{GL}_2$ cases

**Double elliptic $\mathrm{GL}_2$ case.** Consider the $\mathrm{GL}_2$ example explicitly. Rewrite expression (3.2) for the Lax operator of the elliptic Ruijsenaars-Schneider model in terms of the Kronecker function (A.12):

$$\hat{L}_{ij}^{RS}(z, \eta, \hbar) = \frac{\vartheta(-\eta)}{\vartheta'(0)}\Phi(z, q_i - q_j - \eta)\prod_{k\neq j}\frac{\vartheta(q_{jk} + \eta)}{\vartheta(q_{jk})}e^{\hbar\partial_j}, \tag{C.1}$$

so that for $N = 2$ we have

$$\hat{L}^{\text{RS}}(z, \eta, \hbar) = \frac{\vartheta(-\eta)}{\vartheta'(0)} \begin{pmatrix} \Phi(z, -\eta) \frac{\vartheta(q_{12}+\eta)}{\vartheta(q_{12})} e^{\hbar\partial_1} & \Phi(z, q_{12}-\eta) \frac{\vartheta(q_{21}+\eta)}{\vartheta(q_{21})} e^{\hbar\partial_2} \\ \Phi(z, q_{21}-\eta) \frac{\vartheta(q_{12}+\eta)}{\vartheta(q_{12})} e^{\hbar\partial_1} & \Phi(z, -\eta) \frac{\vartheta(q_{21}+\eta)}{\vartheta(q_{21})} e^{\hbar\partial_2} \end{pmatrix}. \tag{C.2}$$

After the substitution into (3.3) one needs to calculate the expression

$$\Phi(z, -k_1\eta)\Phi(z, -k_2\eta) - \Phi(z, q_{21}-k_1\eta)\Phi(z, q_{12}-k_2\eta) \overset{(A.15)}{=}$$

$$= \Phi(z, -(k_1+k_2)\eta)\Big(E_1(q_{12}+k_1\eta) + E_1(q_{21}+k_2\eta) - E_1(k_1\eta) - E_1(k_2\eta)\Big). \tag{C.3}$$

The expression in the brackets is simplified via (A.16), and the function $\Phi(z, -(k_1+k_2)\eta)$ should be substituted in terms of theta functions (A.12). This yields the answer (2.26) for $N = 2$.

**Dual to the elliptic Ruijsenaars-Schneider $\text{GL}_2$ case.** The Lax matrix $\hat{L}^{\text{RS}}$ of the trigonometric Ruijsenaars-Schneider model (3.10) is as follows:

$$\hat{L}^{\text{RS}} = \begin{pmatrix} \frac{tx_1-x_2}{x_1-x_2} q^{x_1\partial_1} & \frac{(1-t)x_2}{x_1-x_2} q^{x_2\partial_2} \\ \frac{(1-t)x_1}{x_2-x_1} q^{x_1\partial_1} & \frac{tx_2-x_1}{x_2-x_1} q^{x_2\partial_2} \end{pmatrix}. \tag{C.4}$$

Then the matrix $\hat{\mathcal{L}}(\lambda)$ (2.45) takes the form:

$$\hat{\mathcal{L}}(\lambda) = \sum_{n\in\mathbb{Z}} \omega^{\frac{n^2-n}{2}}(-\lambda)^n \begin{pmatrix} \frac{t^n x_1-x_2}{x_1-x_2} q^{nx_1\partial_1} & \frac{(1-t^n)x_2}{x_1-x_2} q^{nx_2\partial_2} \\ \frac{(1-t^n)x_1}{x_2-x_1} q^{nx_1\partial_1} & \frac{t^n x_2-x_1}{x_2-x_1} q^{nx_2\partial_2} \end{pmatrix}. \tag{C.5}$$

Computing its determinant one obtains:

$$\hat{\mathcal{O}}(\lambda) = \det\hat{\mathcal{L}}(\lambda) = \sum_{(n_1,n_2)\in\mathbb{Z}\times\mathbb{Z}} \omega^{\frac{n_1^2-n_1}{2}+\frac{n_2^2-n_2}{2}}(-\lambda)^{n_1+n_2} \times \tag{C.6}$$

$$\times \frac{(t^{n_1}x_1-x_2)(t^{n_2}x_2-x_1) - (1-t^{n_1})(1-t^{n_2})x_1x_2}{(x_1-x_2)(x_2-x_1)} q^{n_1x_1\partial_1+n_2x_2\partial_2}$$

$$= \sum_{(n_1,n_2)\in\mathbb{Z}\times\mathbb{Z}} \omega^{\frac{n_1^2-n_1}{2}+\frac{n_2^2-n_2}{2}}(-\lambda)^{n_1+n_2} \frac{t^{n_1}x_1-t^{n_2}x_2}{x_1-x_2} q^{n_1x_1\partial_1+n_2x_2\partial_2},$$

as it should be according to (1.1).

**Dual to the elliptic Calogero-Moser $\text{GL}_2$ case.** In the $N = 2$ case the Lax matrix $\hat{L}^{\text{RS}}$ of the rational Ruijsenaars-Schneider model (3.16) is as follows:

$$\hat{L}^{\text{RS}} = \begin{pmatrix} \frac{z-\eta}{\eta} \frac{q_{12}+\eta}{q_{12}} e^{\hbar\partial_1} & \frac{\eta(z-\eta+q_{12})}{z\,q_{21}} e^{\hbar\partial_2} \\ \frac{\eta(z-\eta+q_{21})}{z\,q_{12}} e^{\hbar\partial_1} & \frac{z-\eta}{\eta} \frac{q_{21}+\eta}{q_{21}} e^{\hbar\partial_2} \end{pmatrix}. \tag{C.7}$$

Then the matrix $\hat{\mathcal{L}}(\lambda)$ (2.45) takes the form:

$$\hat{\mathcal{L}}(\lambda) = \sum_{k\in\mathbb{Z}} \omega^{\frac{k^2-k}{2}}(-\lambda)^k \begin{pmatrix} \frac{z-k\eta}{k\eta} \frac{q_{12}+k\eta}{q_{12}} e^{k\hbar\partial_1} & \frac{\eta(z-k\eta+q_{12})}{z\,q_{21}} e^{k\hbar\partial_2} \\ \frac{\eta(z-k\eta+q_{21})}{z\,q_{12}} e^{k\hbar\partial_1} & \frac{z-k\eta}{k\eta} \frac{q_{21}+k\eta}{q_{21}} e^{k\hbar\partial_2} \end{pmatrix}. \tag{C.8}$$

Computing its determinant one gets (3.17) for $N = 2$.

## 11 Appendix D: Relation between $\mathcal{O}'(z,\lambda)$ and $\mathcal{O}(\lambda)$

The operators $\mathcal{O}'_n$ are generated by the function:

$$\hat{\mathcal{O}}'(z,\lambda) = \sum_{k \in \mathbb{Z}} \frac{\vartheta(z - k\eta)}{\vartheta(z)} \lambda^k \hat{\mathcal{O}}'_k =$$

$$= \sum_{n_1,\dots,n_N \in \mathbb{Z}} \frac{\vartheta(z - \eta \sum_{i=1}^N n_i)}{\vartheta(z)} \omega^{\sum_i \frac{n_i^2 - n_i}{2}} (-\lambda)^{\sum_i^N n_i} \prod_{i<j}^N \frac{\vartheta(q_i - q_j + \eta(n_i - n_j))}{\vartheta(q_i - q_j)} \prod_i^N e^{\hbar n_i \partial_{q_i}}.$$

(D.1)

They are related to the operators $\mathcal{O}_n$ generated by

$$\hat{\mathcal{O}}(\lambda) = \sum_{n_1,\dots,n_N \in \mathbb{Z}} \omega^{\sum_i \frac{n_i^2 - n_i}{2}} (-\lambda)^{\sum_i n_i} \prod_{i<j}^N \frac{\theta_p(t^{n_i - n_j} \frac{x_i}{x_j})}{\theta_p(\frac{x_i}{x_j})} \prod_i^N q^{n_i x_i \partial_i} = \sum_{n \in \mathbb{Z}} \lambda^n \hat{\mathcal{O}}_n,$$

(D.2)

in the following way:

$$\mathcal{O}'_n = h^{-1} \mathcal{O}_n h,$$

(D.3)

where

$$h = h(x_1,\dots,x_N) = \prod_{i<j} \left( \frac{x_i}{x_j} \right)^{\frac{\eta}{2\hbar}}.$$

(D.4)

This follows from the relation between the two theta functions (A.4).

## 12 Appendix E: Easier proof of the main theorems 2.1 and 2.3

In this Appendix we give one more proof of our main theorems, using the presentation, which is very close to the book [48].

*Proof of theorem 2.1:* Denote:

$$\Delta(x) = \prod_{i<j} (x_i - x_j).$$

Let us expand the determinant:

$$\frac{1}{\Delta(x)} \sum_{\sigma \in S_N} (-1)^{|\sigma|} x^{\sigma\delta} \prod_{i=1}^N \theta_\omega(\lambda t^{(\sigma\delta)_i} q^{x_i \partial_i}) =$$

$$= \frac{1}{\Delta(x)} \sum_{\sigma \in S_N} (-1)^{|\sigma|} x^{\sigma\delta} \prod_{i=1}^N \left( \sum_{n_i \in \mathbb{Z}} (-\lambda)^{n_i} \omega^{\frac{n_i^2 - n_i}{2}} (t^{(\sigma\delta)_i})^{n_i} q^{n_i x_i \partial_i} \right).$$

Gathering all terms in front of the fixed power of $u$ one obtains:

$$= \sum_{n_1,\dots,n_N \in \mathbb{Z}} (-\lambda)^{\sum_i n_i} \omega^{\sum_i \frac{n_i^2 - n_i}{2}} \frac{1}{\Delta(x)} \sum_{\sigma \in S_N} (-1)^{|\sigma|} x^{\sigma\delta} \prod_{i=1}^N (t^{(\sigma\delta)_i})^{n_i} \prod_{i=1}^N q^{n_i x_i \partial_i}.$$

The multiplication by $\prod_{i=1}^N (t^{(\sigma\delta)_i})^{n_i}$ could be expressed as a conjugation of $\Delta(x)$ by the shift operator $\prod_{i=1}^N t^{n_i x_i \partial_i}$:

$$\sum_{n_1,\dots,n_N \in \mathbb{Z}} (-\lambda)^{\sum_i n_i} \omega^{\sum_i \frac{n_i^2 - n_i}{2}} \frac{1}{\Delta(x)} \left( \prod_{i=1}^N t^{n_i x_i \partial_i} \right) \sum_{\sigma \in S_N} (-1)^{|\sigma|} x^{\sigma\delta} \left( \prod_{i=1}^N t^{n_i x_i \partial_i} \right)^{-1} \prod_{i=1}^N q^{n_i x_i \partial_i} =$$

$$= \sum_{n_1,\dots,n_N \in \mathbb{Z}} (-\lambda)^{\sum_i n_i} \omega^{\sum_i \frac{n_i^2 - n_i}{2}} \prod_{i<j} \frac{t^{n_i} x_i - t^{n_j} x_j}{x_i - x_j} \prod_{i=1}^N q^{n_i x_i \partial_i}. \quad \blacksquare$$

*Proof of theorem 2.3:* We need to do the same thing as in the proof of theorem 2.1 but for the determinant:

$$\frac{1}{\det \Xi_{ij}(q_j,z)} \det_{1 \leq i,j \leq N} \left\{ \sum_{n \in \mathbb{Z}} (-\lambda)^n \omega^{\frac{n^2-n}{2}} \Xi_{ij}(q_j + n\eta, z) e^{n\hbar \partial_{q_j}} \right\} =$$

$$= \frac{1}{\det \Xi_{ij}(q_j,z)} \det_{1 \leq i,j \leq N} \left\{ \sum_{k \in \mathbb{Z}} \Xi_{ij,k}(z) e^{(\alpha k + \sigma_{ij})q_j} \theta_\omega (\lambda e^{\alpha k \eta + \sigma_{ij}\eta} e^{\hbar \partial_{q_j}}) \right\}$$

$$= \frac{1}{\det \Xi_{ij}(q_j,z)} \sum_{\sigma \in S_N} (-1)^{|\sigma|} \prod_{i=1}^{N} \left[ \sum_{k_i \in \mathbb{Z}} \Xi_{\sigma(i)i,k}(z) e^{(\alpha k_i + \sigma_{\sigma(i)i})q_i} \theta_\omega (\lambda e^{(\alpha k_i + \sigma_{\sigma(i)i})\eta} e^{\hbar \partial_{q_i}}) \right]$$

$$= \frac{1}{\det \Xi_{ij}(q_j,z)} \sum_{\sigma \in S_N} (-1)^{|\sigma|} \prod_{i=1}^{N} \left[ \sum_{k_i \in \mathbb{Z}} \sum_{n_i \in \mathbb{Z}} (-\lambda)^{n_i} \omega^{\frac{n_i^2 - n_i}{2}} \Xi_{\sigma(i)i,k}(z) e^{(\alpha k_i + \sigma_{\sigma(i)i})q_i} e^{(\alpha k_i + \sigma_{\sigma(i)i})n_i \eta} e^{n_i \hbar \partial_{q_i}} \right]$$

$$= \frac{1}{\det \Xi_{ij}(q_j,z)} \sum_{k_1, \dots k_N \in \mathbb{Z}} \sum_{n_1, \dots, n_N \in \mathbb{Z}} (-\lambda)^{\sum_i n_i} \omega^{\sum_i \frac{n_i^2 - n_i}{2}} \times$$

$$\times \sum_{\sigma \in S_N} (-1)^{|\sigma|} \Xi_{\sigma(i)i,k}(z) e^{(\alpha k_i + \sigma_{\sigma(i)i})q_i} \prod_{i=1}^{N} e^{(\alpha k_i + \sigma_{\sigma(i)i})n_i \eta} e^{n_i \hbar \partial_{q_i}},$$

by the same arguments as in the proof of theorem 2.1 above, this expression could be rewritten as:

$$= \frac{1}{\det \Xi_{ij}(q_j,z)} \sum_{n_1, \dots, n_N \in \mathbb{Z}} (-\lambda)^{\sum_i n_i} \omega^{\sum_i \frac{n_i^2 - n_i}{2}} \left( \prod_{i=1}^{N} e^{n_i \eta \partial_{q_i}} \right) \det \Xi_{ij}(q_j,z) \left( \prod_{i=1}^{N} e^{n_i \eta \partial_{q_i}} \right)^{-1} \prod_{i=1}^{N} e^{n_i \hbar \partial_{q_i}}.$$

By evaluating the determinant $\det \Xi_{ij}(q_j,z)$ via (B.13), one obtains:

$$= \sum_{n_1, \dots, n_N \in \mathbb{Z}} (-\lambda)^{\sum_i n_i} \omega^{\sum_i \frac{n_i^2 - n_i}{2}} \frac{\vartheta(z - Nq_0 - \eta \sum_i n_i)}{\vartheta(z - Nq_0)} \prod_{i=1}^{N} \frac{\vartheta(q_i - q_j + \eta n_i - \eta n_j)}{\vartheta(q_i - q_j)} \prod_{i=1}^{N} e^{n_i \hbar \partial_{q_i}} =$$
$$= \hat{\mathcal{O}}'(z - Nq_0, \lambda). \qquad \blacksquare$$

## Acknowledgments

We are grateful to G. Aminov, A. Grosky, A. Mironov, A. Morozov, V. Rubtsov, I. Sechin, Sh. Shakirov and especially to Y. Zenkevich for useful comments and discussions. The work of A. Zotov is supported by the Russian Science Foundation under grant 19-11-00062 and performed in Steklov Mathematical Institute of Russian Academy of Sciences.

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
