# Peer review of "Characteristic determinant and Manakov triple for the double elliptic integrable system"

_SciPost Physics, doi:SciPost Phys. 10, 055 (2021)_

## Round 1 · Referee Report · Anonymous · 2021-1-4

Report
Double elliptic integrable system is a unique integrable system, since both coordinates and momenta take values
in two (different) elliptic curves. In the paradigm of the correspondence of the integrable system and supersymmetric
gauge theory, the classical double elliptic system is associated with the Seiberg-Witten curve of six dimensional
Yang-Mills theory, where one of the elliptic curves parametrizes the coupling constant and the other comes from
a compactification from six to four dimensions. The problem of quantization of double elliptic system is interesting
from the view point of integrable system. But it will also shed light on the still mysterious six dimensional
superconformal gauge theory.
Recently a quantum version of the double elliptic system is proposed by Koroteev and Shakirov.
The present article clarifies some aspects of the proposal.
In my opinion the most interesting achievement is an exact and simple formula (Theorem 4.1)
for the generating function of eigenvalues of the commuting dual Ruijsenaar Hamiltonians.
The result is obtained as a byproduct of a determinant representation for the generating function
of the Hamiltonians of the double elliptic system. (Technically the definition of the generating function
is modified from the original formulation by Koroteev and Shakirov so that the derivation becomes easy.)
A key ingredient for the derivation is the IRF-Vertex correspondence which is one of the most intriguing
results in the theory of the quantum elliptic algebra.
In this paper some aspects of the classical counterpart are also presented. The appearance of the Manakov triple
representation of the Lax operator seems significant. Unfortunately I cannot fully appreciate possible implications
underlying this observation. But in total I can recommend this paper for publication.

---

## Round 1 · Referee Report · Yegor Zenkevich · 2021-2-2

Strengths
1) The paper contains novel identities for the quantum and classical Hamiltonians of the double elliptic (or Dell) integrable system, which lies at the very top of the standard rational-trigonometric-elliptic classification of integrable models. All simpler models such as Ruijsenaars-Schneider and Calogero are its limiting cases. At least for this reason any progress on this very general (and complicated) case is of high value.
2) The paper introduces an averaging procedure, which turns Ruijsenaars-Schneider Lax matrix into a "Lax" matrix related to the Dell system. This procedure is very natural and might be applicable elsewhere.
3) All the theorems proven in the paper are very constructive. A lot of explicit expressions are given. Those will certainly be useful in further investigations related to the Dell systems.
Weaknesses
1) The grammar in some sections of the paper is not up to standard. I have indicated the sentences which I find confusing in the "Style" section of requested changes.
2) Some notations are unclear. These are also listed in the requested changes in the "Style" section.
3) Of course one would dream about finding an honest Lax pair for the classical (or even quantum) Dell system --- not a Manakov triple. However currently this is probably too much to ask for.
Report
The results presented in the paper are valid, new and interesting. I think that the paper is generally well-structured and clearly written, though there are some grammatical mistakes, a few confusing notations and typos. I also have a couple of technical questions for the authors, which however do not affect the main message and results of the paper. After these minor issues are resolved I would definitely recommend the paper to be published in SciPost Physics.
Requested changes
Subject:
1) p.5, Eq. (1.5): it is stated after Eq. (1.1) that the operators O_n are non-commuting. Therefore, the generating functions \mathcal{O}(u) should not in general commute for different values of u. How comes they still have joint eigenvalues E(u)_{\lambda}? If this happens only in the limit p -> 0 this should be written more clearly.
2) p.9, beginning of sec. 2.1.2: I understand the technical reason for introducing an extra spectral parameter z in the generating function O'(\lambda, z). Still I have a general question: why does one need to introduce z if the number of "independent" coefficients of the generating function remains the same?
3) p.9, Theorem 2.3: Do I understand correctly that any Stackel type matrix with determinant (2.27) gives the same generating function (2.28), (2.29)? If this is true then probably this should be made more clear in the text. Alternatively, if the theorem is true only for \Xi of the form (B.6), (B.9), (B.12) this should also be made clear.
4) p.10, Eq.(2.31): the signs in front of q_0 seem to be off: Eq.(B.5) defines \bar{q}_j = q_j - q_0, while Eq.(2.31) implies that q_j = \bar{q}_j - q_0. Which expression is correct?
Style and notations:
1) p.3, last sentence: "left ordered products" --- perhaps this can be made less ambiguous by giving an example. Do the indices of the terms in the product increase from left to right or vice versa?
2) p.5, first line: "A key property of these matrices, which will use, ..." should perhaps read "which we will use".
3) p.6, first paragraph of sec.2: The second to sixth sentences don't make grammatical sense. Perhaps some of them should be joined into one or (better) the whole paragraph should be rewritten.
4) p.7 Theorem 2.1, p.8 Theorem 2.2: The notation Xi_{i,j} is introduced but not used in the statement of the theorem.
5) p.7,8 Theorem 2.1 and 2.2: To avoid confusion it might me better to introduce separate notation for the generating functions of the Hamiltonians in the trigonometric and rational limits. Currently the letter \hat{\mathcal{O}} denotes three different generating functions --- the original elliptic one (1.1), its trigonometric and rational limits.
6) p.10 after Eq.(2.42): "In order to witness it in its convenient form..." --- perhaps it is better to use another verb, e.g. "rewrite", "recast", "put into".
7) p.12, Eq.(3.6) and after: The argument q_i-q_j of the Lax matrix \hat{L}^{RS}(z,q_i-q_j, n_j \eta, n_j \hbar) does not appear in its definition (3.2). Is \hat{L}^{RS}(z,q_i-q_j, \eta, \hbar) the same as \hat{L}^{RS}(z, \eta, \hbar) from (3.2)? If this is the case, then the notation should be made more uniform. Otherwise the difference between the two matrices should be made more clear. An additional reason why the notation with q_i-q_j in the argument looks confusing is that the matrix element L_{ij} is in fact not a function of q_i-q_j only but of all coordinates q_1,..., q_N.
8) Titles of sec. 3.2, 3.3, 4: it is not immediately obvious what "dual to" mean in the context. If the duality is the p-q duality perhaps this should be indicated.
9) p.15, second line: "let us calculate its image" --- whose? The previous sentence is about the space of all m_{\lambda}, not just one.
10) p.17, before Eq.(5.13): "as was mention in" --- should read "mentioned".
11) p.25, Eq.(A.11): perhaps w_j should read w_i in the r.h.s. Otherwise it is unclear what value should the index j take --- it does not appear in the l.h.s.
12) p.26, Eq.(B.5): q should probably read q_j in the r.h.s.

---

## Round 3 · Referee Report · Yegor Zenkevich (Referee 2) · 2021-2-20

Report

Two typos still need to be corrected (see "Requested changes"). Otherwise, the paper is ready for publication.

Requested changes

(All page, equation and section numbers are according to the new version of the manuscript)

1) p.6, fourth sentence of sec.2: "paring" should read "pairing".

2) p.10, Eq.(2.32): the sign in front of q_0 hasn't been changed and still looks wrong to me. This can also be noticed by comparing Eq.(2.28) to (2.33): the two equations are compatible if there is z - N q_0 instead of z + N q_0 in the last equality of Eq.(2.32).

---

## Round 3 · Author Response

This version of the paper is very close to original one.

---

## Round 3 · List of Changes

We added Appendix E, which contains alternative proofs of main statements. Also some typos were corrected and some comments added. We tried to make all corrections from the lists of recommendations by Reviewers.

---

## Editorial Decision

published